# EDIF: Editing via Dynamic Interactive Tuning with Feedback

## Abstract

Although text-guided image editing (TIE) has advanced rapidly, most prior works remain object-centric and rely on attention maps or masks to localize and modify specific objects. In this paper, we propose a method of *Editing via Dynamic Interactive Tuning* (EDIF) that adaptively trades off source-image structure and instruction fidelity in difficult scene-centric editing settings. Unlike object editing, scene-centric editing is challenging because the target cannot be clearly localized, and edits need to preserve global structure. To cope with the limitation of TIE systems that typically use a unified conditioning signal and ignore the block-wise variation in the internal behavior of the model, we show that inside the model, the source-image condition and the text-prompt embedding act with layer-dependent directions and strengths. We also demonstrate both empirically and theoretically that the editing state can be diagnosed using the source image signal-to-noise ratio and VLM logits, which indicate whether the edited image faithfully reflects the intended editing prompt. By constructing a Pareto line between these two objectives, EDIF adaptively modulates the source-image and editing-text conditions, guiding each denoising step to stay close to this line for balanced optimization. Extensive experiments on ImgEdit, EmuEdit-Bench, and Places365 show that EDIF achieves state-of-the-art performance in various scene-editing scenarios, including indoor and outdoor environments.

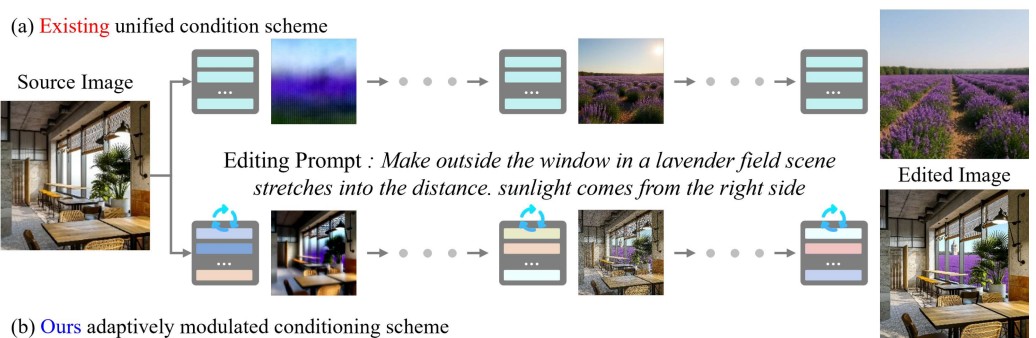

Figure 1: Comparison of edited images of EDIF from existing methods. Contrary to existing methods that rely on unified editing condition and fail to balance structural preservation and semantic alignment, EDIF exploits feedback signals (SNR and VLM logits) to monitor the editing pathway and block-wise adaptively adjust editing condition at each step. This per-step adaptation enables faithful and reliable edits.

## 1 Introduction

Recent advances in rectified-flow transformers have driven substantial progress in both image generation and editing (Yang et al., 2024; Lipman et al., 2023; Labs et al., 2025). Existing editing approaches, such as inversion-based (Rout et al., 2024; Wang et al., 2025), attention-based (**?**), mask-guided (Couairon et al., 2022; Yu et al., 2023) and latent-based (Shuai et al., 2024) methods, focus primarily on object-centric editing. However, scene-centric

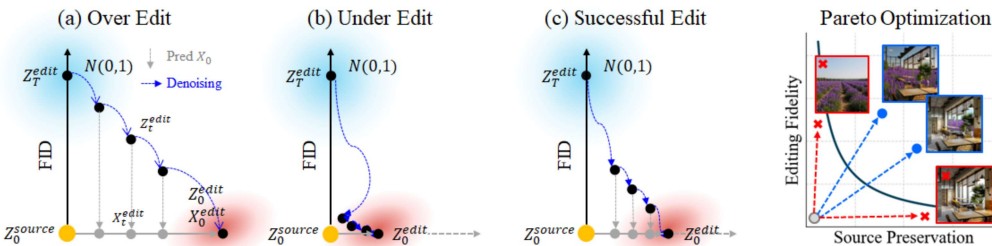

Figure 2: Conceptual visualization of the editing pathway and EDIF's Pareto-guided control. (Left) (a) over-editing, (b) under-editing, and (c) successful editing. (Right) the Pareto line that EDIF targets to achieve balanced, successful edits.

editing cannot localize specific regions, making existing approaches difficult to apply. Prior scene-centric editing approaches attempt to work out this issue by training on large curated datasets (Labs et al., 2025; Brooks et al., 2023), but this strategy is time-consuming and has a low generalization.

As shown in Figure 2, the editing process can be categorized into three regimes: over-edit, under-edit, and successful edit. For a successful edit, a balance between source preservation and prompt fidelity is essential. Existing methods apply the condition strength across the entire network uniformly, which is suboptimal for maintaining this balance. Instead, we argue that adaptively modulating the condition strengths across blocks is necessary to achieve this balance. To this end, we draw a Pareto line defined by the two objectives of source preservation and editing fidelity and update condition strengths according to the current editing state, thereby guiding each denoising step to converge closer to this line. In particular, the preservation of the source is captured by the signal-to-noise ratio (SNR) referenced by the source ($\text{SNR}_{\text{src}}$), while the fidelity of the editing is measured by VLM logits.

SNR refers to the signal-to-noise ratio that quantifies the relative strength of the desired signal compared to background noise. SNRsrc, derived from this idea, quantifies how much information from the source image remains in the latent denoised. Empirically, latents on a successful editing trajectory deviate less from the source than those from over-edited cases, resulting in consistently higher $\text{SNR}_{\text{src}}$. Therefore, we adopt $\text{SNR}_{\text{src}}$ as a reliable indicator of the denoising state. VLM logits measure semantic agreement with the editing instruction, allowing us to monitor whether the latent representations along the editing pathway align with the editing prompt.

To determine how to control the editing pathway, we conduct blockwise ablation experiments by selectively removing text or image conditioning from individual transformer blocks of the model. Interestingly, we find that removing the condition on specific blocks can counter-intuitively improve both source preservation and editing fidelity. This observation demonstrates that the editing pathway can be effectively controlled through adaptive blockwise adjustment of conditioning. Building on these insights, we propose EDIF, a feedback-driven framework that addresses the dual objectives of source preservation and prompt fidelity through stepwise Pareto optimization.

We extensively evaluate EDIF on ImgEdit-Bench (Ye et al., 2025), Emu Edit Bench (Sheynin et al., 2023) and a Places365 based (Zhou et al., 2016) dataset with GPT-4o generated prompts (Zhou et al., 2016) and demonstrate that EDIF consistently outperforms prior methods in achieving a superior structure semantics trade-off and delivers more robust, faithful scene-centric edits.

## 2 RELATED WORK

**Text Instruction-based Image Editing**  Text instruction-based image editing (TIE) has evolved through multiple approaches: attention-based (Fluxspace (Dalva et al., 2024), FreeFlux (Wei et al., 2025b), MasaCtrl (Cao et al., 2023), Prompt-to-Prompt (Hertz

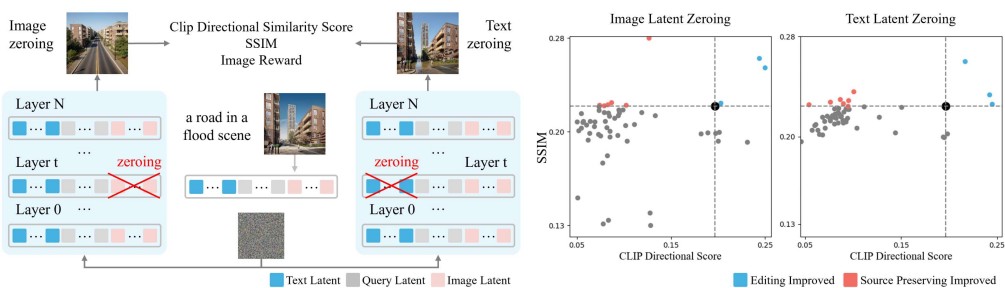

Figure 3: Blockwise image and text condition zeroing experiment. We selectively zero out either the image condition latents or the text condition latents within individual transformer blocks to assess their respective contributions to image editing. The left column shows results when image latents are zeroed and the right column shows results when text latents are zeroed.

et al., 2023), Stable Flow (Avrahami et al., 2025)), inversion-based ( (Rout et al., 2024), RFEdit (Wang et al., 2025), and FlowEdit (Kulikov et al., 2025)), mask-guided (DiffEdit (Couairon et al., 2022), FlexEdit (Nguyen et al., 2024), Inpaint Anything (Yu et al., 2023), Mag-Edit (Mao et al., 2023), SDEdit (Meng et al., 2022), UltraEdit (Zhao et al., 2024), and Flux-text (Lan et al., 2025)). While these strategies show strong performance in object-centric editing, they face fundamental limitations in scene-centric editing, where modification and preservation must occur at the same spatial location. The development of scene-centric editing has mainly relied on training approaches such as Instruct-Pix2Pix (Brooks et al., 2023), RefEdit (Pathiraja et al., 2025), InstructDiffusion (Geng et al., 2023), MagicBrush (Zhang et al., 2024), OmniEdit (Wei et al., 2025a), FLUX.1 Kontext (Labs et al., 2025). However, such training-based methods suffer from limited generalization, as they require training whenever new domains or editing types are introduced, making them highly inefficient in practice.

**Multi-Objective Optimization**   In multi-objective optimization, a solution is considered Pareto optimal if no objective can be improved without causing at least another objective to deteriorate. Such solutions represent the best possible trade-offs among conflicting goals. Recent methods such as PCGrad (Yu et al., 2020), CAGrad (Liu et al., 2024), and Nash-MTL (Navon et al., 2022) explicitly aim to find solutions on the Pareto front, while others like GradNorm (Chen et al., 2018) balance gradients across tasks to indirectly mitigate conflicts. PROUD (Yao et al., 2024) treats multi-objective generation jointly—rather than optimizing each property independently—by steering samples to lie on the Pareto front of conflicting objectives. Likewise, ParetoFlow (Yuan et al., 2025) guides flow-based sampling to approximate the Pareto front.

## 3 Method

### 3.1 Motivation

Our study focuses on Kontext (Labs et al., 2025), a DiT-based image editing model that in a transformer block, processes source image and text condition independently. Kontext employs two types of transformer blocks: 19 dual-stream blocks and 38 single-stream blocks (Peebles & Xie, 2023).

In diffusion models which composed of CNN based U-Nets, blockwise roles have been extensively investigated (Si et al., 2023; Li et al., 2023). By contrast, DiT-based architectures is completely composed of transformer block, which is fundamentally different architecture. The blockwise behavior remains comparatively underexplored.

To study blockwise functions in Kontext, we adopt an ablation-driven analysis. Using Chat-GPT (OpenAI, 2024), we generated $N$=10 editing prompts for 30 source images, preparing

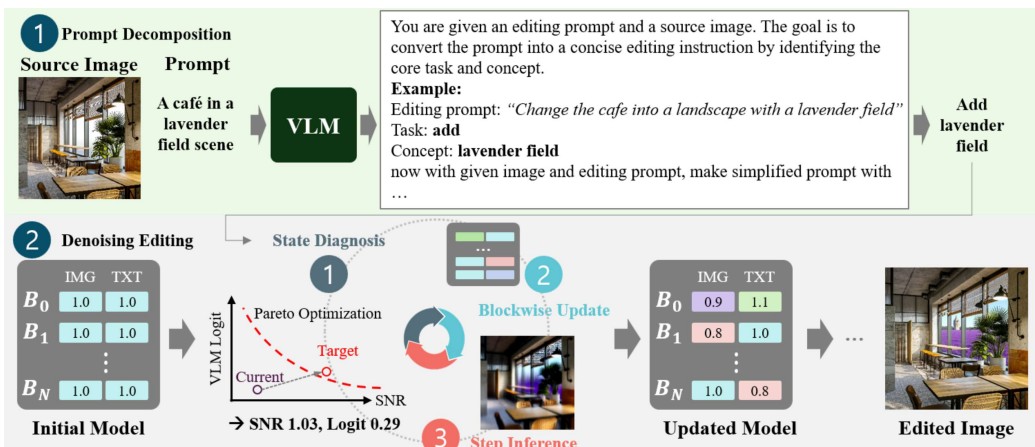

Figure 4: The feedback driven EDIF pipeline is composed of 2 stages. In stage 1, a VLM decomposes the user free-form prompt into key concepts, used to diagnose the editing latent. Stage 2 proceeds in three steps: (1) run an initial inference and measure the source-image SNR and VLM logits, (2) compare the state to the Pareto line and control condition strength adaptively, and (3) inference with block-wise adapted model. This loop repeats until the state enters the Pareto region, and the diffusion denoiser updates the latent.

a total of $N{=}300$ editing tasks. For each transformer block in Kontext, $b \in \mathcal{B}$, we performed two types of zeroing experiments, as illustrated in Figure 3. We evaluated three metrics: structural preservation via SSIM (Sara et al., 2019), instruction adherence via CLIP Directional Similarity (Gal et al., 2021), and perceptual quality via ImageReward (Xu et al., 2023a).

The results, plotted in Figure 3, show that zeroing image condition or text condition in certain blocks significantly increased the original image structure or editing performance, while in some blocks it causes the degradation of image quality. These results are counter-intuitive: one might expect that weakening the source-image condition would harm source preservation, and zeroing the text-prompt condition would degrade instruction fidelity. However, our experiments show that, depending on the layer, zeroing can in fact be beneficial. Based on these findings, we performed a complementary amplification test following the same protocol. Rather than zeroing, we scale the condition strength by $\times 2$ at a single block. Doubling image condition sometimes yields improved editing, while in text condition doubling experiment, there is no noticable improvement.

This experiment yields two results. First, layers play distinct roles, and within each layer, image and text conditioning behave differently. Second, instead of applying uniform conditioning across layers, we tailor conditioning blockwise, which yields more reliable edits. Guided by our ablation, we partition blocks into six groups: four under image conditioning—(G1) lowering aids structure preservation, (G2) lowering aids editing, (G3) scaling up aids structure preservation, and (G4) scaling up aids editing—and two under text conditioning—(G5) lowering aids structure preservation and (G6) lowering aids editing.

## 3.2 EDITING STATE DIAGNOSIS

As shown in Figure 2, editing results vary by sample. This implies that the conditions must be adjusted per sample according to its current editing state. To monitor the state of editing throughout denoising, we analyze two diagnostics (1) whether the source information $x_0$ is preserved and (2) whether the instruction $p$ is faithfully reflected.

**SNR Signal** Given source image $x_0$, one goal of editing is to preserve structure of $x_0$. Conventional SNR measures the amount of noise contained in an image. We extend this definition and propose the source-based SNR. Given source image $x_0$, $y_t$ is the latent along the editing pathway. We aim to measure how much of the signal of $x_0$ is present in $y_t$.

$$\text{SNR}_{\text{src}}(y_t, x_0) = 10 \cdot \log_{10} \left( \frac{\sum_p x_0(p)^2 + \varepsilon}{\sum_p \left( x_0(p) - y_t(p) \right)^2 + \varepsilon} \right). \tag{1}$$

where $p$ indexes pixel locations and $\varepsilon$ is a small constant for numerical stability. Unlike conventional SNR, which compares the current latent to the ground-truth clean image, our extended source SNR compares the source image to the target image.

A higher $\text{SNR}_{\text{src}}$ indicates that the comparison latent retains more of the source image signal. Experimentally, we observe that successful editing trajectories consistently exhibit higher $\text{SNR}_{\text{src}}$ than failed ones (see Supplementary C.2). It indicates that this signal is a reliable diagnostic of structural fidelity during the denoising process.

**VLM Logits**  Given editing prompt $c$, another goal of editing is for $c$ to be perceptible in $\hat{x}_0$. To achieve this, as shown in Figure 2, $c$ must also be present in $y_t$. How can we determine whether the current state is properly following $c$? Conventionally, CLIP Directional Similarity ($\text{CLIP}_{\text{dir}}$) is used to assess editing success. However, it evaluates the final clean edited image and thus does not reflect the state of latents along the editing pathway, where substantial noise remains. In other words, $\text{CLIP}_{\text{dir}}$ is ill-suited for diagnosing intermediate, noisy latents.

We adopt a VLM-based alternative to $\text{CLIP}_{\text{dir}}$ for diagnosing intermediate states. We query the VLM to assess whether $y_t$ follows the instruction. Empirically, even though $y_t$ is in a noisy state, VLM logits are more stable than $\text{CLIP}_{\text{dir}}$ : the CLIP-based score often fluctuates across time-steps and proves unstable as a feedback signal, VLM logits remain consistent and reliable, making them better suited for guiding editing. Therefore, we use VLM logits to verify whether the latent accurately follows the prompt or not. (see Supplementary Section D.3). Here, VLM logits refers not to a binary *yes/no* output but to the softmax score for the *yes* response.

### 3.3 EDIF PIPELINE

Scene-centric image editing is a multi-object task that preserves the structure of the source image $x_0$ while applying a scene-change editing prompt $c$. We frame these two objectives on a Pareto frint and, during editing, diagnose progress using the source SNR $\text{SNR}_{\text{src}}$ and VLM logits. If the current state falls outside the Pareto region, we adjust the image and text condition strengths blockwise. Following iterative diffusion denoising, this procedure yields the final edited image.

**Prompt Decomposition**  User-provided prompts contain not only target editing concepts but also source image descriptions and redundant details, which can obscure the intended edit. To enable VLM-based evaluation, we first decompose the prompt into key concepts and compute VLM logits to verify whether the latent in the editing pathway aligns with the prompt.

VLM extracts the core editing concept from each sentence and adding *add* or *make* as task with key concept. For example, the free-form prompt 'change this cafe into a lavender field' is transformed into the key command *add lavender field*. Compared to the original user prompt, the reduced key concept prompt, composed solely of essential editing concepts, was experimentally verified to be more appropriate to evaluate whether the latent was well edited in ablation 4.5.

**Denoising Step**  To assess whether the structure of $x_0$ is preserved in $y_t$, and whether changes occur according to $c$, we use $\text{SNR}_{\text{src}}$ and VLM logit. Accurate diagnosis requires that $y_t$ be properly configured; therefore, we first obtain the predicted clean image.

$$\hat{z}_0' = \hat{z}_t - f_\theta(\hat{z}_t, t, \mathbf{z}_0, \mathbf{z}_c),$$

which follows directly from the DiT-based rectified-flow objective. Here, $f_\theta$ denotes the rectified flow–matching model, and $\mathbf{z}_0$ denotes the source-image latent. However, $\hat{x}'_0$ (the decoded version of $\hat{z}'_0$) can be noisy depending on the denoising timestep. When noise is high, dual signals can be unreliable. To obtain a cleaner proxy, we compare $\hat{x}'_0$ with a pixel space reconstruction $\hat{x}'_t$ from the current latent (i.e., decoded from $\hat{z}_t$) and choose the one with the higher SNR. (Further details are provided in the Supplementary Section D.1.)

We then compute $\mathrm{SNR}_{\mathrm{src}}$ and the VLM logits on $y_t$ for diagnosis. EDIF compares the current state against the Pareto front. If it falls outside the Pareto region, it blockwise adaptively updates the strengths of the image and text conditions following our zeroing experiment. More concretely, when the SNR deviates substantially from the Pareto line, we scale up the conditioning in G3 and scale down in G1 and G5. When the VLM logit deviates, we scale up in G4 and scale down in G2 and G6, thus adapting conditioning per block. Using the updated model, we then infer at the current timestep and iterate this procedure until the $\mathrm{SNR}_{\mathrm{src}}$ and VLM logits approach the Pareto front. (For details on the model update, see Figure 4.) Through this iterative process, EDIF achieves edits that satisfy the multi-objective criteria.

## 4 EXPERIMENTS

### 4.1 EXPERIMENTAL SETUP

**Comparison models.** We compare EDIF against TIE methods spanning diffusion and transformer based. These include Stable Diffusion v1.5-based (InstructDiffusion Geng et al. (2023), MagicBrush Zhang et al. (2024), InfEdit Xu et al. (2023b), InstructPix2Pix Brooks et al. (2023))(IP2P), SD3-based RefEdit Pathiraja et al. (2025), SDXL-based OmniEdit Wei et al. (2025a), and Step1X-Edit Liu et al. (2025), which leverages VLM guidance, and our baseline Kontext Labs et al. (2025), which is based on Flux (Kang et al., 2025)

**Dataset.** We evaluate EDIF on two public benchmarks and one scene-editing dataset that we constructed. For benchmarks, we use the scene-related portions of ImgEdit Bench (Ye et al., 2025) and Emu Edit Bench (Sheynin et al., 2023), filtering instructions that request *global/background/style* changes. For a deeper scene-focused analysis, we derive a scene-centric set from Places365 (Zhou et al., 2016). We sample 100 validation images and automatically generate 20 edit instructions per image with GPT-4o, yielding 2,000 image–instruction pairs.

**Evaluation Protocol.** For ImgEdit Bench we follow its protocol that uses vision–language models to score instruction following preservation and quality. Following the evaluation protocol of Emu Edit Bench, we report $\mathrm{CLIP}_{\mathrm{dir}}$ and $\mathrm{CLIP}_{\mathrm{out}}$ evaluate compliance with the editing instruction and $\mathrm{CLIP}_{\mathrm{img}}$ measure structural preservation of the source image to the edited image. We also compute Image Reward (ImgRWD) (Xu et al., 2023a) to assess the quality of the edited image. In addition, we evaluate with VIE-Score (Ku et al., 2024) which reports Semantic Consistency (SC), reflecting how well the edit follows the prompt and Perceptual Quality (PQ) that captures naturalness.

### 4.2 EXPERIMENTS ON IMGEDIT-BENCHMARK

Table 1 summarizes the performance on ImgEdit-Bench. MagicBrush, InfEdit, IP2P, and RefEdit obtain relatively low PQ scores, indicating limited visual authenticity and naturalness. In contrast, Step1X and Kontext achieve a higher PQ, producing more naturally-looking results. For the VLM-based evaluation, we use Qwen2.5-VL (Yang et al., 2025) to score the output. Step1X and Kontext achieve high quality scores but show somewhat lower source-structure preservation. EDIF, while slightly below Kontext in quality, achieves higher preservation with more balanced between editing fidelity and source preservation.

Figure 5 provides qualitative evidence. For the *snowy* instruction, InstructDiff, InfEdit, CosXL, and Step1X-Edit do not convincingly convey winter characteristics. MagicBrush, RefEdit, and Kontext often break the scene structure, producing edits that diverge from the source. Only IP2P and our method follow the instructions with preserving the source. However, IP2P transfers fine details too literally, resulting in unnatural outputs.

| Data | | ImgEdit-Bench | | | | | | | | | | |
| Method | Base | VLM-Based | | | VIE Score | | Metric-Based | | | | | |
| | | Instruct↑ | Preserve↑ | Quality↑ | SC↑ | PQ↑ | CLIP$_{dir}$↑ | CLIP$_{out}$↑ | CLIP$_{img}$↑ | SSIM↑ | ImgRWD↑ | FID↓ |
|---|---|---|---|---|---|---|---|---|---|---|---|---|
| Instruct-Diff | SD1.5 | 3.840 | 3.900 | 2.992 | 5.165 | 5.035 | 0.109 | 0.078 | 0.700 | 0.825 | 0.271 | 274.959 |
| MagicBrush | SD1.5 | 4.120 | 3.510 | 3.524 | 4.913 | 4.994 | 0.272 | 0.140 | 1.232 | 0.725 | 0.235 | 275.239 |
| InfEdit | SD1.5 | 3.810 | 3.990 | 3.001 | 4.844 | 4.844 | 0.182 | 0.320 | 0.709 | 0.656 | 0.371 | 274.729 |
| IP2P | SD1.5 | 3.680 | 3.910 | 3.113 | 4.829 | 4.992 | 0.108 | 0.240 | 0.821 | 0.641 | 0.240 | 274.799 |
| RefEdit-SD3 | SD3 | 4.020 | 4.010 | 3.374 | 5.117 | 4.967 | 0.278 | **0.340** | 1.082 | 0.929 | 0.689 | 275.139 |
| CosXL | SDXL | 4.240 | **4.310** | 3.624 | **5.544** | 5.065 | 0.311 | 0.209 | 0.971 | 0.873 | 0.701 | 203.586 |
| Step1X-Edit | DiT+VLM | 4.240 | 3.620 | **4.220** | 4.852 | 5.071 | 0.301 | 0.210 | 1.928 | 0.664 | 0.319 | 275.359 |
| Kontext | Flux | 4.292 | 3.559 | 4.121 | 4.925 | **5.075** | 0.389 | 0.259 | 0.863 | 0.641 | 0.813 | **194.558** |
| EDIF | Flux | **4.319** | 3.707 | 4.115 | 5.179 | 5.050 | **0.410** | 0.228 | **1.990** | **0.991** | 0.836 | 275.438 |

| | Emu Edit Bench | | | | | | Places365 | | | | | |
| Method | CLIP$_{dir}$↑ | CLIP$_{out}$↑ | CLIP$_{img}$↑ | ImgRWD↑ | SC↑ | PQ↑ | CLIP$_{dir}$↑ | CLIP$_{out}$↑ | CLIP$_{img}$↑ | ImgRWD↑ | SC↑ | PQ↑ |
|---|---|---|---|---|---|---|---|---|---|---|---|---|
| Instruct-Diff | 0.131 | 0.157 | 0.754 | 0.152 | 4.832 | 3.971 | 0.101 | 0.127 | 0.730 | 0.123 | 4.102 | 4.012 |
| MagicBrush | 0.193 | 0.205 | 0.854 | 0.164 | 4.832 | 3.971 | 0.120 | 0.110 | 0.854 | 0.120 | 4.832 | 4.847 |
| InfEdit | 0.209 | 0.295 | 0.788 | 0.056 | 5.053 | 4.255 | 0.175 | 0.140 | 0.788 | 0.104 | 5.053 | 5.058 |
| IP2P | 0.185 | 0.280 | 0.787 | 0.104 | 4.902 | 4.085 | 0.280 | 0.060 | 0.857 | 0.112 | 4.902 | 4.961 |
| RefEdit-SD3 | 0.180 | 0.203 | 0.765 | 0.121 | 5.005 | 4.114 | 0.303 | 0.300 | 0.765 | 0.198 | 5.005 | 4.878 |
| CosXL | 0.210 | 0.289 | **0.824** | 0.197 | 5.227 | 4.112 | 0.120 | 0.118 | 0.824 | 0.120 | 5.227 | 4.801 |
| Step1X-Edit | 0.215 | **0.297** | 0.803 | 0.241 | **5.332** | 4.104 | 0.227 | 0.295 | 0.710 | 0.149 | 5.012 | 5.042 |
| Kontext | 0.240 | 0.288 | 0.801 | **0.247** | 4.210 | **4.451** | 0.358 | 0.279 | 0.701 | **0.211** | 4.877 | **5.138** |
| EDIF | **0.260** | 0.292 | 0.821 | 0.244 | 5.110 | 4.415 | **0.381** | **0.310** | **0.911** | 0.201 | **5.750** | 5.501 |

Table 1: The experimental results are reported on three datasets. The first table presents the results on ImgEdit-Bench, the left side of the second table shows the results on EmuEdit-Bench, and the third table reports the results on Places365.

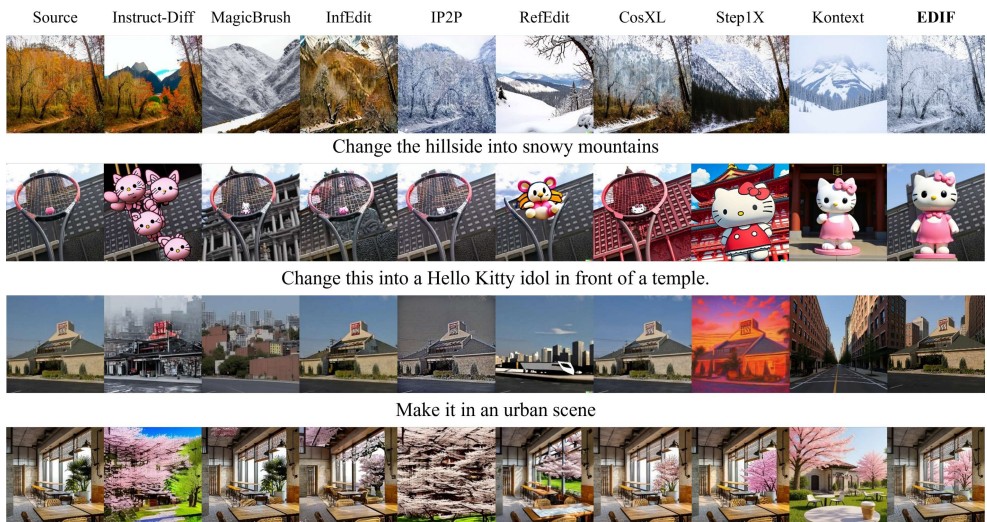

Figure 5: Visualization of editing results. The leftmost column shows the source image; subsequent columns show model outputs. From top to bottom, rows correspond to the editing result on ImgEdit-Bench, HQ-Edit, and Places365 results. The last row show the editing result on indoor scene image.

## 4.3 EXPERIMENTS ON EMUEDIT

While the overall trends on EmuEdit are consistent with those on ImgEdit-Bench, EDIF shows a distinctive advantage on this benchmark. As illustrated in Figure. 5, no baseline produced a natural transformation for the instruction *Hello Kitty idol in front of a temple*. Most edits either failed to realize the concept or exhibited conspicuous artifacts. Except for Step1X and Kontext, the baselines tended to preserve the original tennis racket shape, resulting in an unsuccessful rendering of the *Kitty idol*. Step1X did realize the idol but hallucinated the background buildings, while Kontext failed to preserve the source content.

In contrast, EDIF successfully executed the instruction, indicating that the feedback-driven procedure can handle this challenging task reliably. In contrast, EDIF followed the text prompt while preserving scene plausibility, yielding a more natural and faithful edit.

**Editing Pathway.** In Figure 6, we compare the editing pathways of EDIF and Kontext (Labs et al., 2025). While Kontext exhibits over-editing even at early denoising steps and ultimately fails to preserve the source, EDIF dynamically adjusts conditioning strength, producing edits that align with the prompt while retaining structural fidelity. EDIF adaptively tunes the conditioning to correct low initial SNR and steer SNR and VLM logits toward the Pareto frontier, achieving a balanced trade-off between semantic alignment and structural preservation.

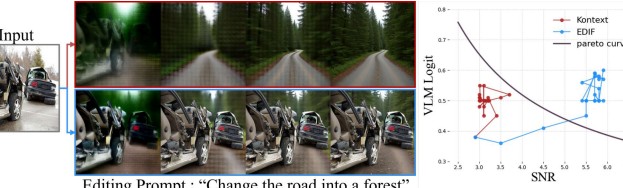

Editing Prompt : "Change the road into a forest"

Figure 6: Editing pathway analysis of Kontext and EDIF. The left side illustrates intermediate images along the denoising trajectory, while the right side shows the corresponding SNR and VLM logit curves over timesteps. Red lines denote Kontext and blue lines denote EDIF.

### 4.4 EXPERIMENTS ON PLACES365

To assess scene-level editing, we additionally evaluate on Places365 within ImgEdit-Bench. Table 1 summarizes the results, and Figure 5 illustrates examples in the third row. InstructDiff exhibits reduced naturalness and noticeably degraded image quality. MagicBrush, RefEdit, and Kontext tend to over-edit, collapsing source structure and producing outputs that diverge from the original content. InfEdit and CosXL often under-edit, resulting in minimal changes to the scene. IP2P and Step1X-Edit frequently deviate from the prompt and fail to deliver

| Method | Strength | CLIP$_{dir}$ ↑ | CLIP$_{out}$ ↑ | CLIP$_{img}$ ↑ | ImgRWD↑ |
|--------|----------|-----------|-----------|-----------|---------|
| Uniform | 0.3 | 0.310 | 0.332 | 0.877 | 0.892 |
| | 0.5 | 0.275 | 0.320 | 0.928 | 0.929 |
| | 1 | 0.276 | 0.317 | 0.932 | 0.911 |
| | 2 | 0.276 | 0.331 | 0.925 | 0.937 |
| | 3 | 0.274 | 0.307 | 0.910 | 0.893 |
| Dynamic | | 0.378 | 0.301 | 1.125 | 0.901 |

Table 2: Scores for fixed and dynamic latent strength scaling.

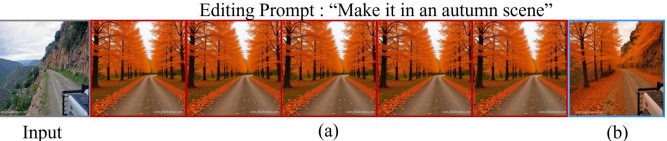

Editing Prompt : "Make it in an autumn scene"

Input      (a)      (b)

Figure 7: The first column shows the source image. (a) Result when a fixed strength is uniformly applied. (b) Result from EDIF with dynamic adjustment.

balanced edits. In contrast, our EDIF preserves the core structure while following the urban-scene instruction effectively. These observations align with results on the other datasets, where Kontext, MagicBrush, and RefEdit sacrifice preservation due to aggressive edits, CosXL and InfEdit retain too much of the original, and InstructDiff and Step1X-Edit yield somewhat awkward modifications.

**Conditioning Control Strategy.** EDIF dynamically adjusts the condition strength at each inference timestep, whereas CFG uses a single global scale. We evaluate two settings on 20 randomly sampled Places365 images (Zhou et al., 2016) with the same prompts as in the main Places365 experiments. First, we apply a fixed scaling factor uniformly to text embeddings across all layers, sweeping it from 0.3 to 3.0. Second, the EDIF case, where adaptively controls the strength as inference.

Table 2 and Figure 7 show the results. When a fixed scaling factor is applied uniformly across layers, the editing performance degrades, producing low CLIP$_{dir}$ and CLIP$_{out}$. In contrast, adapting the scaling factor per block leads to successful edits. These experiments

empirically demonstrate that layers have distinct roles, and therefore, uniform scaling cannot satisfy the trade-off between original structure preservation and editing fidelity.

**User Study**   We conducted a user study to further assess the quality of editing. A total of 20 source images were sampled, and participants were asked to evaluate four outputs per case (the original image and edits produced by three models). Each comparison included three questions: (1) structural consistency, (2) prompt fidelity, and (3) naturalness of the edit. Although CosXL performs relatively well in structural preservation, it often struggles to produce faithful edits. EDIF effectively maintains the structure of the source image while following the editing instructions, whereas Kontext tends to over-edit and shows lower structural consistency. Refer to Appendix E.2 for experiment details.

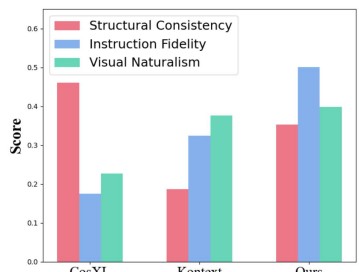

Figure 8:   Result of the user study comparing three models.

## 4.5 Complex Editing Prompt

**Pareto-Line Construction.** SNR-based feedback is an iterative tuning method aimed at the Pareto front. To examine how the construction of the Pareto frontier affects performance, we conduct experiments in which the structural threshold $\tau$ on the Pareto frontier is varied from 1.0 to 4.0.

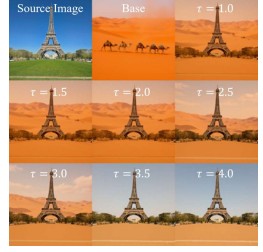 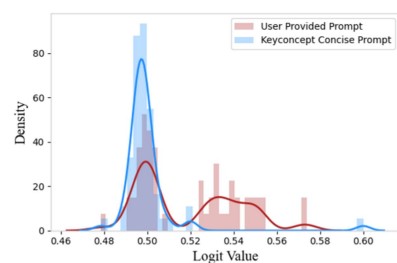

Figure 9: Ablation study: (a) effect of Pareto-line construction, and (b) precise prompt effeict in VLM feedback.

Figure. 9 shows the result. Without feedback, structural integrity collapses and edits fail. With a small threshold such as $\tau$=1.0, the overall layout of the source image is preserved, and as $\tau$ increases, structural preservation increases steadily. $\tau$ indeed acts as a practical control of the condition strength.

**Key Concept Extraction.**   We compare VLM feedback using the decomposed reduced key-concept prompts against raw free-form prompts. As shown in Figure 9, raw free-form prompts often restate attributes of the source image, which can artificially inflate the *yes* logits even when little or no editing actually occurs. In contrast, the key-concept prompt provides stricter judgments, resulting in lower *yes* logits. This yields more robust evaluations across various user prompts, while also producing logits that are more stable throughout the denoising process.

## 5 Conclusion

We introduce EDIF, a feedback driven algorithm for scene-centric image editing. Given a source image and a textual editing prompt, EDIF set a Pareto line for editing two objectives of preserving the source image structure and achieving prompt fidelity. Along the denoising trajectory, EDIF diagnoses the latent at every step, checking whether the source signal is retained and whether the prompt semantics are faithfully expressed. Following the state, EDIF adaptively adjusts transformer conditioning to steer the editing trajectory onto, and keep it within, the Pareto line. Unlike prior approaches that apply identical controls uniformly across all layers, EDIF performs blockwise control whose strength scaling are adapted based on the trajectory's position relative to the Pareto line. We provide a theoretical analysis of the diagnostic signals that drive these control decisions, and we empirically show that the procedure is robust. Quantitative and qualitative evaluations on ImgEdit, EmuEdit, and Places365 demonstrate state-of-the-art performance across diverse scene-editing tasks.

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
