# EDIF: Editing via Dynamic Interactive Tuning with Feedback Supplementary Material

This supplementary material presents:

- A. Limitations.

- B. Future Research.

- C. Motivation Details of Experiments and Architecture Analysis.

- D. Detailed Explanation on Singals.

- E. Experiments Details and Qualitative Results

## A    LIMITATIONS

Our model has been developed and evaluated in the Flux-based Kontext framework (Labs et al., 2025), where image and text are treated as independent conditions. While this setting provides a clean and controllable environment, it also limits the ability to explore more integrated or joint conditioning strategies. Beyond pure generation, recent LLM-guided image editing methods (Wu et al., 2025) highlight the need to evaluate broader conditioning strategies across diverse architectures.

## B    FUTURE RESEARCH

From our investigation, we obtained a key insight regarding condition control. In the conventional CFG (Ho & Salimans, 2022), the condition guidance scale $s$ is applied uniformly across the network and cannot be modulated selectively. However, our study demonstrates that when conditions are applied in a layer-wise instead of in a uniform manner, controlled editing becomes feasible. We believe this opens a new direction for effectively managing editing conditions. Currently, fine-grained control in editing models relies on drag-based methods (Mou et al., 2023; Bertazzini et al., 2025) or training additional target-specific concepts (Gandikota et al., 2023). Through our approach, we show that an effective TIE-based methodology can also enable such fine-grained control. In future research, we plan to broaden the scope of controlled editing beyond existing paradigms.

## C    ZEROING EXPERIMENT

### C.1    BLOCKWISE ZEROING AND ATTENTION REDISTRIBUTION.

Blockwise analyses of U-Net–based models have been reported (Ho et al., 2020; Si et al., 2023; Li et al., 2023; Kim et al., 2025). In contrast, in DiT-based models (Peebles & Xie, 2023), blockwise characterization has not yet been extensively explored. Among these, StableFlow (Avrahami et al., 2025) emphasizes that different layers exhibit distinct properties through a layer analysis of DiT models. Building on these prior studies, we aim to provide a more systematic investigation of the blockwise behavior in DiT-based models.

To examine the role of each condition across model layers, we apply blockwise latent zeroing, where either the image or text latent is set to zero at individual transformer blocks during inference. Our experiments are conducted on 30 source images paired with 10 carefully designed editing prompts, resulting in a total of 300 editing tasks. Zeroing is applied separately to image and text latents across all 57 transformer blocks in FLUX (Yang et al., 2024).

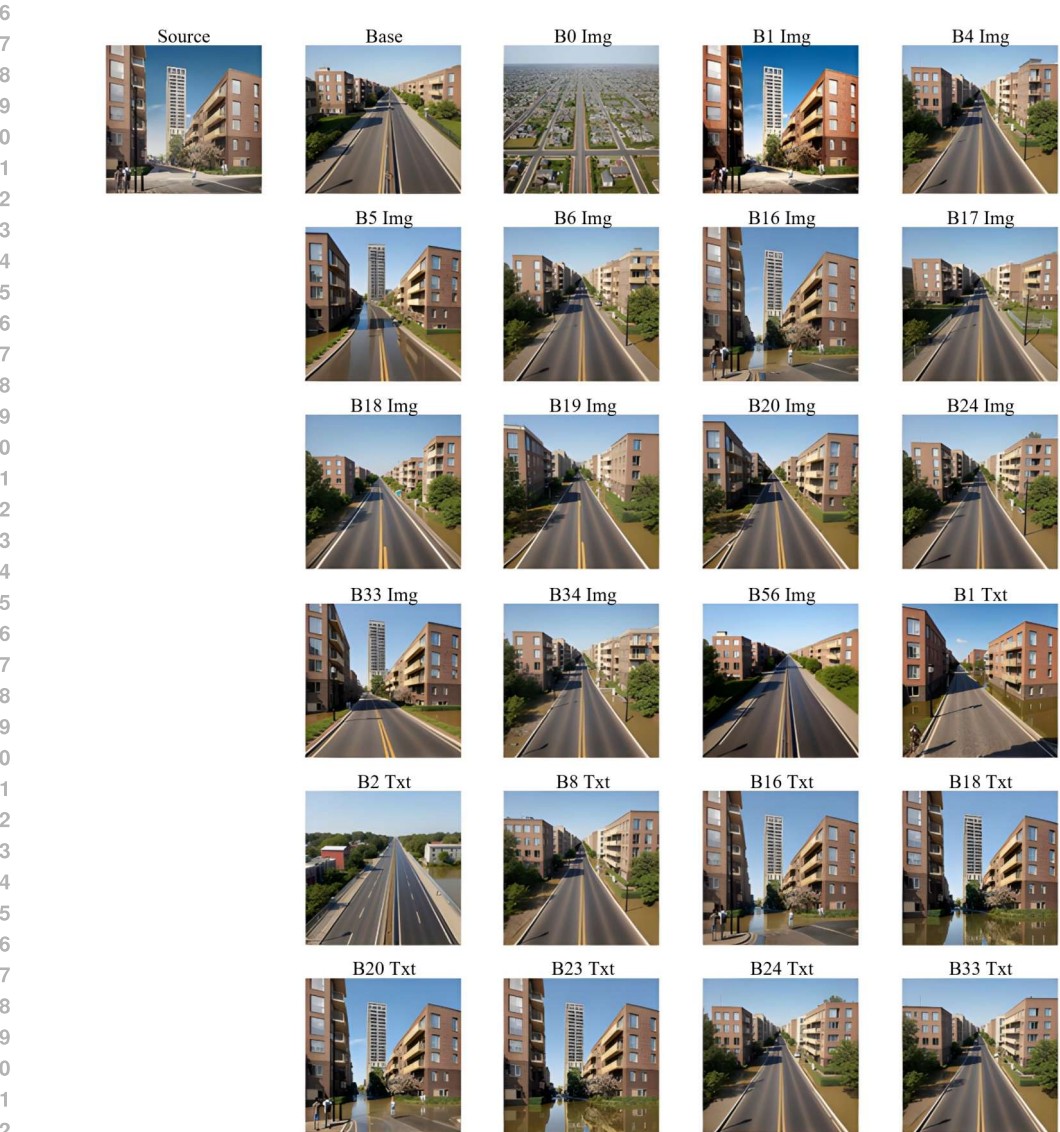

Figure C1: Effect of blockwise zeroing of image and text condition latents in the Kontext. The title of each image in this plot specifies the block index and the latent type that was zeroed. For example, *B18 Image* indicates that the image latent at block 18 was set to zero.

The edited images are then evaluated using CLIP directional similarity (Gal et al., 2021), which measures whether the edits align with the given prompt, SSIM (Wang et al., 2004), which evaluates the preservation of structure from the source image, and ImageReward (Xu et al., 2023a), which assesses the overall quality of the generated images. As shown in Figure 3, zeroing the image latent at certain blocks can unexpectedly increase the CLIP directional score or SSIM compared to the base model. Although the effect is generally smaller for text-zeroing than for image-zeroing, we also observe that zeroing out text latent at some blocks improves the CLIP directional score.

The example of results are shown in Figure C1. Figure C1 presents the source image and edited outputs using the prompt *a road in a flood scene.* The base model fails to render any visible *flood.* Interestingly, zeroing either the image or text conditioning latent at specific blocks can actually improve performance: Zeroing the image conditioning at blocks 5, 16, and 33, or the text conditioning at blocks 16, 18, 20, and 23, restores the source structure.

Zeroing the image conditioning at blocks 5, 6, 16, 17, 18, 19, 20, 24, and 43, or the text conditioning at blocks 2, 8, 16, 18, 20, 23, 24, 33, and 55, reveals the intended flood edits that were absent under the default configuration.

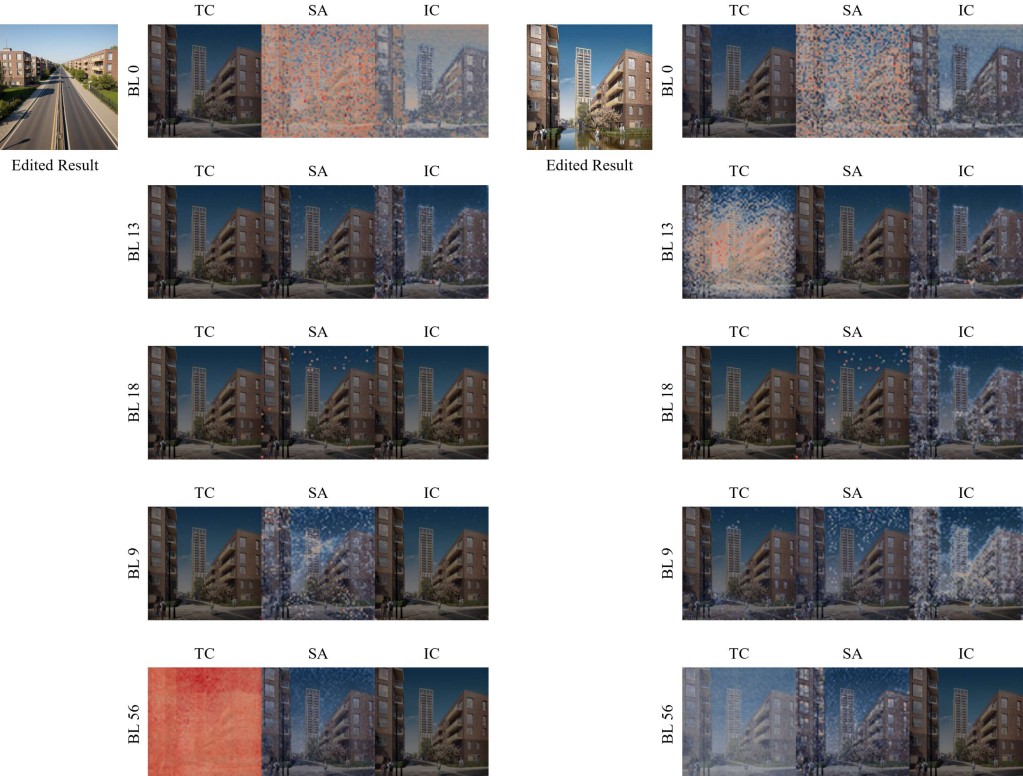

Figure C2: Comparison between the baseline (left) and the case where the text latent is zeroed at different blocks (right). For each image panel, the label on the left indicates the block at which the corresponding attention was probed (e.g., BL0 denotes Block 0). TC, SA, and IC denote text cross-attention, self-attention, and image cross-attention, respectively.

To understand why disabling conditioning at a specific block influences the entire image, we analyze per-block attention maps for each zeroing intervention. In Figure C2, TC, SA, and IC denote text cross-attention, self-attention, and image cross-attention, respectively.

As shown in Figure C2, zeroing at Block 18 redistributes attention that was previously over-concentrated in self-attention. At Block 13, attention to the text condition increases, while at Block 18, attention to the source image increases. At Block 9, regions previously focused solely on self-attention shift toward the source image, while at Block 56, attention that was excessively concentrated on the text in the base setting spreads to other components.

Intuitively, one might expect that reducing the text latent would decrease TC. However, when zeroing one block, the effect is not confined to that block alone. Adjusting a single block does not remain confined to that block alone, and text zeroing does not necessarily result in a simple reduction of attention to the text. Adjustments at a specific block propagate to others, and the condition change evolves in ways that contradict simple intuition.

## C.2 ARCHITECTURAL ANALYSIS

Contrary to intuition, our zeroing experiments reveal that the conditions governing image editing are organically intertwined, and that manipulating one condition can induce qualitatively different outcomes through its interaction with the others. We argue this in-

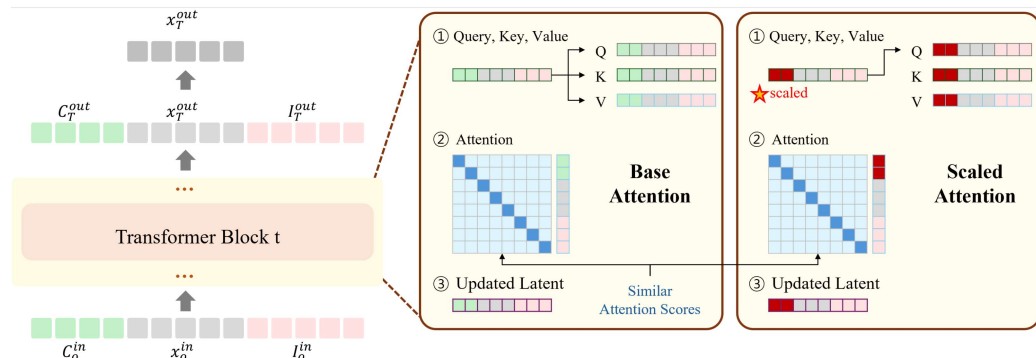

Figure C3: Attention at specific blocks in Kontext. The left panels show attention under the base setting, while the right panels show attention after scaling the text embeddings. Despite scaling, the resulting attention scores remain broadly similar to the base.

tertwined influences to the underlying model architecture. To clarify this point, we present an architectural analysis of DiT-based editing models.

A autoencoder encodes the source image into tokens $z_0$, which are appended to the target-image tokens $z_t$ to form a long visual sequence $[z_t, z_0]$. Within each transformer block, this visual sequence is concatenated with the text embeddings $z_c$ to yield $[z_c; z_t; z_0]$. The resulting sequence is then used as the queries, keys, and values for self-attention.

Consider a transformer block $b^\star$ where we scale the text tokens by a factor $s$. This modification affects the text embeddings of $Q$, $K$, and $V$ at that block and thereby perturbs the attention. However, because attention is composed of vector products followed by a softmax normalization, the resulting attention distributions remain close to the base routing pattern. In this regime, scaling neither erases the text information nor induces large deviations from the original text signal.

Instead, it redistributes attention between the text and source embeddings. This departs from the intuitive expectation that weakening the text embedding should simply diminish text-driven editing. Because the cross-attention patterns are highly similar, scaling does not behave as intuitively expected when compared to the baseline; decreasing the scaling does not reliably reduce text semantics, nor does increasing it consistently strengthen semantics. Rather, attenuation can instead rebalance the text and source image condition and result in improved edited outcomes.

# D   DETAILED EXPLANATION ON SIGNALS

## D.1   COMPARISON LATENT SELECTION

According to the flow-matching framework, the noisy latent at timestep $t \in [0,1]$ can be written as a convex combination of the clean image and Gaussian noise:

$$z_t = \sigma_t \epsilon + (1 - \sigma_t) z_0, \qquad \epsilon \sim \mathcal{N}(0, I). \tag{D1}$$

where $z_0$ is the latent of the clean image. The flow-matching training objective is

$$\mathcal{L}_\theta = \mathbb{E}_{t \sim p(t), x, \mathbf{z}_i, \mathbf{z}_c} \left\| f_\theta(z_t, t, \mathbf{z}_0, \mathbf{z}_c) - (\varepsilon - z_0) \right\|_2^2, \tag{D2}$$

Following Eq. D2, the clean latent can be estimated from $z_t$ as

$$\hat{z}_0' = \varepsilon - f_\theta(z_t, t, \mathbf{z}_0, \mathbf{z}_c), \tag{D3}$$

where $f_\theta$ denotes the flow-matching model. However, naive using the predicted clean image $\hat{z}_0'$ at every timestep can yield misleading feedback during denoising.

In theoretical, SNR is

$$\mathrm{SNR}(t) = \frac{(1 - \sigma_t)^2}{\sigma_t^2} \tag{D4}$$

Figure D1 compares two SNR curves: the theoretical (scheduler-derived) one and the SNR computed from the predicted clean image $\hat{x}_0$. As seen in the figure, the $\hat{x}_0$ based one is higher in the early denoising steps; however, as denoising progresses, the theoretical SNR surpasses it. This pattern is also visible in the example images on the right. Early in denoising, the predicted clean image looks relatively clean, but at later steps the decoded current latent shows much clear image than the clean predicted one.

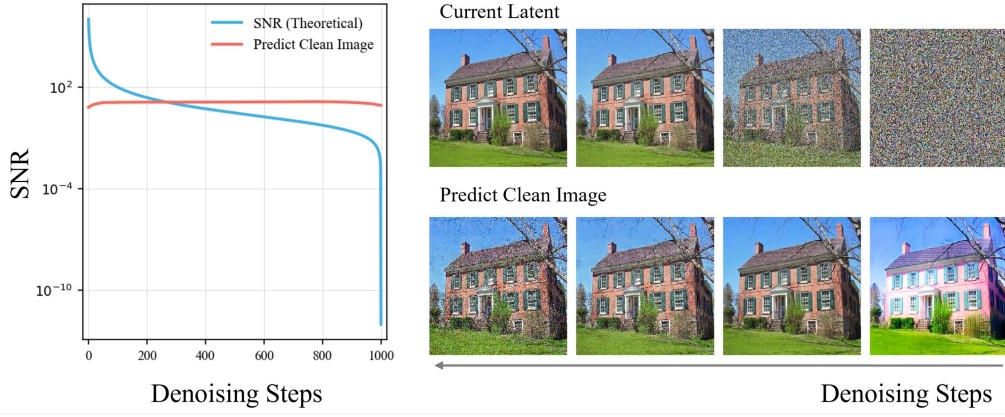

Figure D1: SNR dynamics across two contrasting editing cases. (Left) SNR over denoising steps. The pink curve is computed from the predicted clean image, while the blue curve is the theoretical (scheduler-derived) SNR. (Right) intermediate images along the editing pathway during denoising. The top row shows the decoded current latent, and the row below shows the pixel-space predicted clean image. The rightmost column corresponds to denoising time $t{=}1$, with columns to the left approaching $t{=}0$.

In EDIF, we diagnose the latent state along the editing pathway. At each denoising timestep, we select the latent that provides a cleaner and more reliable signal. As the SNR results

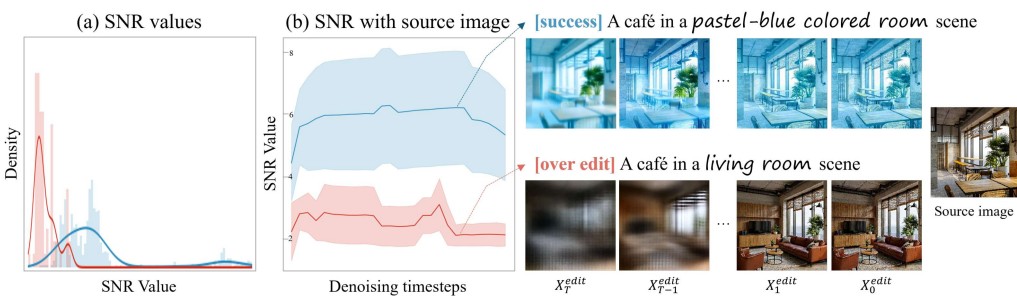

Figure D2: Evolution of $\mathrm{SNR}_{\mathrm{src}}$. (a) shows $\mathrm{SNR}_{\mathrm{src}}$ across all runs, and (b) shows $\mathrm{SNR}_{\mathrm{src}}$ throughout the denoising process. The panels on the right indicate the values obtained in successful editing cases and in failed cases, respectively.

show, the predicted clean image and the decoded current latent are strong candidates; by comparing their SNRs with respect to the source and choosing the one with the higher SNR, we obtain a more accurate estimate of the editing state.

## D.2 SNR in Diffusion Dynamics

The signal-to-noise ratio (SNR) is a fundamental quantity that measures the relative strength of the desired signal to the noise. Formally, the forward diffusion process in DiT is defined as Eq. D1 SNR decreases monotonically along the forward noising process, reflecting the fact that as $t$ increases, the latent $x_t$ becomes increasingly dominated by Gaussian noise. During denoising, the reverse dynamics aim to recover $x_0$ by progressively improving the SNR of the intermediate latent states.

The theoretical SNR follows Eq. D4. In practice, one can compute it directly from the latent as

$$\mathrm{SNR}_{\mathrm{num}}(t) = \frac{\left\| \hat{x}_0^{(t)} \right\|_2^2}{\left\| x_t - \hat{x}_0^{(t)} \right\|_2^2}, \tag{D5}$$

where $\hat{x}_0^{(t)}$ denotes the predicted clean image at step $t$. Intuitively, when $x_t$ is highly corrupted by noise, the discrepancy $\| x_t - \hat{x}_0^{(t)} \|_2^2$ becomes large, yielding low SNR. Although $\mathrm{SNR}_{\mathrm{num}}$ is not identical to the theoretical one, in practice it exhibits the same qualitative trend.

**SNR with Respect to the Original Image** While the previous formulation relies on predicted clean estimates $\hat{x}_0^{(t)}$, Our ultimate question is: how much of the original source image $x_0$ is preserved in the current latent $x_t$? To answer this, we directly compute an SNR that references the ground-truth $x_0$. Specifically, we define

$$\mathrm{SNR}_{\mathrm{src}}(t) = \frac{\| x_0 \|_2^2}{\| x_t - x_0 \|_2^2} \tag{D6}$$

Here, the numerator captures the energy of the original image signal, and the denominator measures the deviation of the current latent from the source. A large value of $\mathrm{SNR}_{\mathrm{src}}(t)$ indicates that the denoised latent closely resembles the source image, while a small value suggests that the latent has diverged significantly due to noise. This formulation effectively quantifies the proportion of source information that remains embedded in the latent representation.

Empirically, we observe that $\text{SNR}_{\text{src}}(t)$ increases monotonically along the denoising trajectory, serving as a reliable indicator of structural preservation with respect to the original image.

Figure D2 presents both quantitative and qualitative evidence that successful edits. (a) When editing succeeds, the SNR is generally higher than in failed cases. Moreover, when stratified by denoising time, the successful editing cases exhibit higher SNR at every timestep. This demonstrates that the source-based SNR is both theoretically reliable and empirically verifiable as a signal, enabling EDIF to leverage it effectively to preserve the structure of the source image.

## D.3 Experiments on VLM Logit

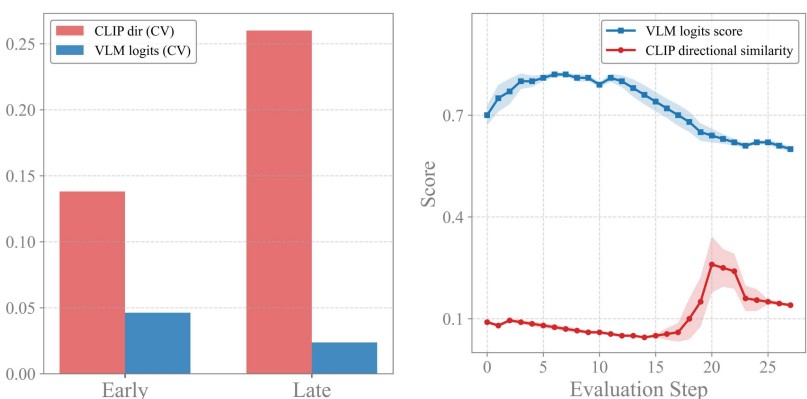

Figure D3: SNR–logit scatter. Blue points denote successful trajectories and red points denote failures. The green curve shows a smoothed Pareto frontier estimated from the successes.

While CLIP directional similarity ($\text{CLIP}_{\text{dir}}$) is widely used as a proxy for editing fidelity, we observed that it can yield inaccurate diagnoses of the editing state—in particular, cases where editing is qualitatively successful yet $\text{CLIP}_{\text{dir}}$ remains spuriously low. To analyze this limitation in greater depth, we compare $\text{CLIP}_{\text{dir}}$ against vision–language model (VLM) logits that directly measure the presence of the target concept.

In our experiment, we used the same image–text prompt pairs as in the zeroing setup, yielding a total of 300 editing tasks. Given an image $x_0$ and a text prompt $c$, we performed editing and measured both $\text{CLIP}_{\text{dir}}$ and VLM logits at each stage. Figure D3 reports the mean and standard deviation (std) of these measurements. The left panel shows the std at early and late denoising steps, where $\text{CLIP}_{\text{dir}}$ exhibits a much larger std than the VLM logits. This indicates that, as denoising proceeds, the VLM logits vary relatively little, whereas $\text{CLIP}_{\text{dir}}$ fluctuates substantially. The right panel corroborates this observation: as shown in the plot, the magnitude of $\text{CLIP}_{\text{dir}}$ is much smaller than that of the VLM logits, which in turn makes its relative changes appear larger and its std higher. In other words, $\text{CLIP}_{\text{dir}}$ is a very small and sensitive quantity, and using such a too-sensitive signal to diagnose the state of the editing latent makes constructing a Pareto line challenging.

We hypothesize that this behavior arises because $\text{CLIP}_{\text{dir}}$ is trained and calibrated primarily on clean, high-quality (low-FID) images. As a result, $\text{CLIP}_{\text{dir}}$ provides unstable signals for noisy intermediate latents along the editing pathway; in such out-of-distribution conditions, the image and text embeddings become overly sensitive to low-level artifacts and stochastic variations.

Moreover, cosine-direction signals are inherently small in scale at intermediate timesteps, which makes constructing a stable Pareto line difficult. Taken together, these factors explain why $\text{CLIP}_{\text{dir}}$ underperforms as a diagnostic metric on noisy latents, even when editing is proceeding successfully.

### D.4 PARETO LINE CONSTRUCTION

We evaluate the same image–prompt pairs used in the zeroing experiments and, during denoising-based editing, record the source-based SNR and the VLM logit at each step from the mid-trajectory to the final step. As visualized in Figure D4, successful edits (blue dots) and failures (red dots) occupy clearly separated regions in the SNR–logit plane. Notably, successful trajectories concentrate within a common region. Connecting the terminal segment of these trajectories yields a Pareto front, as shown in green. This observation motivates casting image editing as a Pareto-style multi-objective optimization problem.

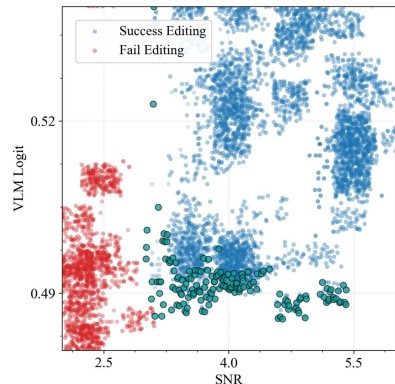

In multi-objective optimization, a solution is Pareto optimal if no objective can be improved without degrading at least one other objective. Applied to image editing, strengthening structure preservation typically weakens prompt fidelity, and vice versa. Hence, successful edits must steer into a coexistence region where both objectives are jointly satisfied. Empirically, the upper envelope of the successful cloud approximates this front. Points on or near this upper bound constitute Pareto-optimal or near-optimal trade-offs.

Figure D4: SNR–logit scatter. Blue points indicate successful trajectories, while red points failures. The green plot denotes the smoothed Pareto line derived from the success case.

While the precise shape of the Pareto line can vary across samples, our experiments consistently show that successful trajectories converge toward this region. This insight motivates the design of EDIF's Pareto-line–based feedback pipeline.

## D.5 EDIF ALGORITHM

We present the pseudocode of EDIF, which (1) selects a comparison latent per step, (2) computes two feedback signals, and (3) applies Pareto-guided blockwise conditioning updates.

---

**Algorithm 1: EDIF (Pareto-guided): Feedback-Driven Image Editing**

---

**Input** : Source image $x_0$; prompt $p$; initial latent $z_T$; decoder $D$;
VLM $\mathcal{M}_{\text{vlm}}$; editing model $\mathcal{M}_{\text{edit}}$; scheduler Sched;
per-block condition scales $\boldsymbol{\kappa}_{\text{img}}^{(t)}, \boldsymbol{\kappa}_{\text{txt}}^{(t)} \in \mathbb{R}^B$;
block-importance masks $\mathbf{m}_{\text{img}}^S, \mathbf{m}_{\text{txt}}^S, \mathbf{m}_{\text{img}}^E, \mathbf{m}_{\text{txt}}^E \in [0,1]^B$;
pareto line $l_p$, pareto line area $R_p$; step sizes $\eta_S$, $\eta_E$; clip bounds $\kappa_{\min}$, $\kappa_{\max}$;
max feedback iterations $K_{\text{fb}}$

**Output:** Edited image $\hat{x}_0$

// Step 1: Prompt distillation (key concept)

1   $p_{\text{key}} \leftarrow \text{DistillPrompt}(x_0, \ p; \ \mathcal{M}_{\text{vlm}})$

2   **for** $t = T \dots 1$ **do**

     // Forward pass with current per-block conditioning

3     $\varepsilon_t \leftarrow \mathcal{M}_{\text{edit}}\big(z_t, \ p, \ x_0; \ \boldsymbol{\kappa}_{\text{img}}^{(t)}, \ \boldsymbol{\kappa}_{\text{txt}}^{(t)}\big)$

4     $\tilde{x}_t \leftarrow D(z_t)$                      // decoded reconstruction

5     $\hat{x}_0^{(t)} \leftarrow \text{CleanEstimate}(z_t, \ \varepsilon_t, \ t, \ p)$       // e.g., scheduler inversion

6     $y^{(t)} \leftarrow \underset{y \in \{\hat{x}_0^{(t)}, \ \tilde{x}_t\}}{\arg\max} \ \text{SNR}_{\text{src}}(y, \ x_0)$

     // Compute feedback signals

7     $S_t \leftarrow \text{SNR}_{\text{src}}(y^{(t)}, \ x_0), \quad E_t \leftarrow \text{VLMLogit}\big(y^{(t)}, \ p_{\text{key}}\big)$

     // Pareto-guided, mask-aware one-sided feedback loop

8     $k \leftarrow 0$

9     **while** $\big(S_t, E_t\big) \in R_p$ **and** $k < K_{\text{fb}}$ **do**

         // Positive deficits to feasible region $R_p$

10       $(\tau_S, \tau_E) \leftarrow \text{ClosestPoint}\big((S_t, E_t), l_p\big)$

11       $e_S \leftarrow 1$ if $\tau_S > S_t$ else $0, \quad e_E \leftarrow 1$ if $\tau_E > E_t$ else $0$

         // One-sided Pareto weights (nonnegative, sum to 1)

12       $d \leftarrow e_S + e_E + 10^{-8}, \quad w_S \leftarrow e_S/d, \quad w_E \leftarrow e_E/d$

         // layerwise updates

13       $\boldsymbol{\kappa}_{\text{img}}^{(t)} \leftarrow \text{clip}\big(\boldsymbol{\kappa}_{\text{img}}^{(t)} + w_S \, \eta_S \, e_S \odot \mathbf{m}_{\text{img}}^S, \ \kappa_{\min}, \ \kappa_{\max}\big)$

14       $\boldsymbol{\kappa}_{\text{txt}}^{(t)} \leftarrow \text{clip}\big(\boldsymbol{\kappa}_{\text{txt}}^{(t)} + w_E \, \eta_E \, e_E \odot \mathbf{m}_{\text{txt}}^E, \ \kappa_{\min}, \ \kappa_{\max}\big)$

         // Re-evaluate with updated conditioning

15       $\varepsilon_t \leftarrow \mathcal{M}_{\text{edit}}\big(z_t, \ p, \ x_0; \ \boldsymbol{\kappa}_{\text{img}}^{(t)}, \ \boldsymbol{\kappa}_{\text{txt}}^{(t)}\big)$

16       $\tilde{x}_t \leftarrow D(z_t), \quad \hat{x}_0^{(t)} \leftarrow \text{CleanEstimate}\big(z_t, \ \varepsilon_t, \ t, \ p_{\text{key}}\big)$

17       $y^{(t)} \leftarrow \underset{y \in \{\hat{x}_0^{(t)}, \ \tilde{x}_t\}}{\arg\max} \ \text{SNR}_{\text{src}}(y, \ x_0)$

18       $S_t \leftarrow \text{SNR}_{\text{src}}(y^{(t)}, \ x_0), \quad E_t \leftarrow \text{VLMLogit}\big(y^{(t)}, \ p_{\text{key}}\big)$

19       $k \leftarrow k + 1$

     // Scheduler step

20     $z_{t-1} \leftarrow \text{Sched}(z_t, \ \varepsilon_t, \ t)$

21   **return** $\hat{x}_0 \leftarrow y^{(t)}$

---

# E  EXPERIMENTS DETAILS

## E.1  EXPERIMENTAL DETAILS ON PLACES365

We conduct large-scale editing experiments on the Places365 images (Zhou et al., 2016) to evaluate generalization across diverse scenes and layouts. We sample 100 validation images from Places365 (Zhou et al., 2016) and, for each source image, generate editing instructions with GPT-4 (OpenAI, 2025) (conditioned on a brief scene description). For every image we create 20 prompts, constrained to semantically plausible edits. Figure E1 illustrates the prompt generation process.

```
### Overview :
You are helping create scene-editing instructions for an image editing model.

## Inputs
SCENE DESCRIPTION: {brief_description_from_image}

## Requirements
Write 3 single-sentence editing instructions that:
- Modify the scene PLAUSIBLY given the description (weather, lighting, materials, minor geometry, time of day).
- Avoid unsafe content, identity targeting, or contradictions with the description.
- Avoid adding implausible objects that cannot reasonably exist in the scene.
- Be concise and concrete; include visual attributes when relevant (lighting/colors/texture/atmosphere).
- Use imperative mood (e.g., "Make the road wet and reflective under soft rain at dusk.").
- Encourage diversity across the 3 outputs (do not repeat the same attribute family).
- Do not mention camera brands or country names unless visually necessary.
- Do not include numbers or labels in the sentence.

### Examples :

# Example Input
SCENE DESCRIPTION: A tidy bedroom with a window, wooden floor, and neutral tones.
# Example Output (JSON Lines)
{"instruction": "Warm the scene with a soft bedside lamp glow and gentle dusk light spilling through the window."}
{"instruction": "Make the wooden floor slightly glossy and enhance the linen bedding texture in a muted gray palette."}
{"instruction": "Shift to an early-morning atmosphere with cool, low-angle light casting long, soft shadows across the floor."}

### Output format (JSON Lines; exactly 3 lines)
For each instruction, output one line of JSON with the field:
{"instruction": "<ONE SENTENCE>"}
{"instruction": "<ONE SENTENCE>"}
{"instruction": "<ONE SENTENCE>"}

Return ONLY these JSON lines. No extra commentary.
```

Figure E1: Example of GPT-based editing prompt generation for Places365 images.

## E.2 USER STUDY DETAILS

To evaluate human preference for scene centric image editing, we conduct a user study capplying our method EDIF with two strong baselines, Kontext (Labs et al., 2025) and CosXL (Wei et al., 2025a). We have recruited 30 participants, including AI researchers and designers. Each trial presented a source image, a natural-language editing prompt, and three edited images labeld with model name of Model A, Model B, and Model C. Participants have ranked the three results from 1st (best) to 3rd (worst) based on: (1) visual quality, (2) structural consistency with the source layout, and (3) alignment with the text instruction. Test samples are drawn from three datasets (ImgEdit-Bench (Ye et al., 2025), Emu Edit Bench (Sheynin et al., 2023), Places365 (Zhou et al., 2016)). Each participant have completed 36 trials. An example question of the user study is provided in Figure E2

### Image editing user study

In this study, you will compare the outputs of three image-editing models. For each trial, a prompt and three corresponding edited images are provided. Your task is to rank the images (1st, 2nd, 3rd) according to overall preference, balancing the following criteria:
- **Visual Quality :** clarity, realism, and level of detail in the edited image.
- **Structural Consistency :** how well the global layout and spatial structure of the source image are preserved after editing.
- **Text-Instruction Alignment :** the degree to which the edit reflects the intended modification described in the prompt.

Images are displayed left to right and labeled A, B, and C. Assign a unique rank (1st, 2nd, 3rd) to each image based on a holistic judgment across all three criteria.

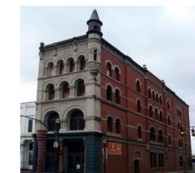

Editing Prompt

Make it into a spring scene

Source Imge

Model A        Model B        Model C

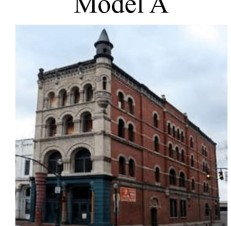 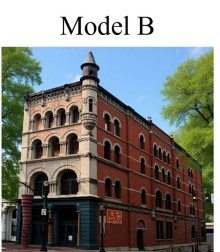 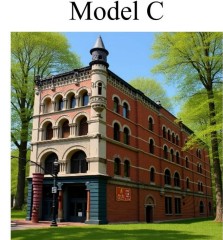

|  | **Model A** | **Model B** | **Model C** |
|---|---|---|---|
| Visual Quality | ○ | ○ | ○ |
| Structure Consistency | ○ | ○ | ○ |
| Text-Instruction Alignment | ○ | ○ | ○ |

Figure E2: Example of user study instructions and a sample question presented to participants.

### E.3 Qualitative Results

We additionally report qualitative results of EDIF from Figure E3 to Figure E6. For a fair comparison, We compare EDIF against a basic Kontext baseline that does not use our algorithm.

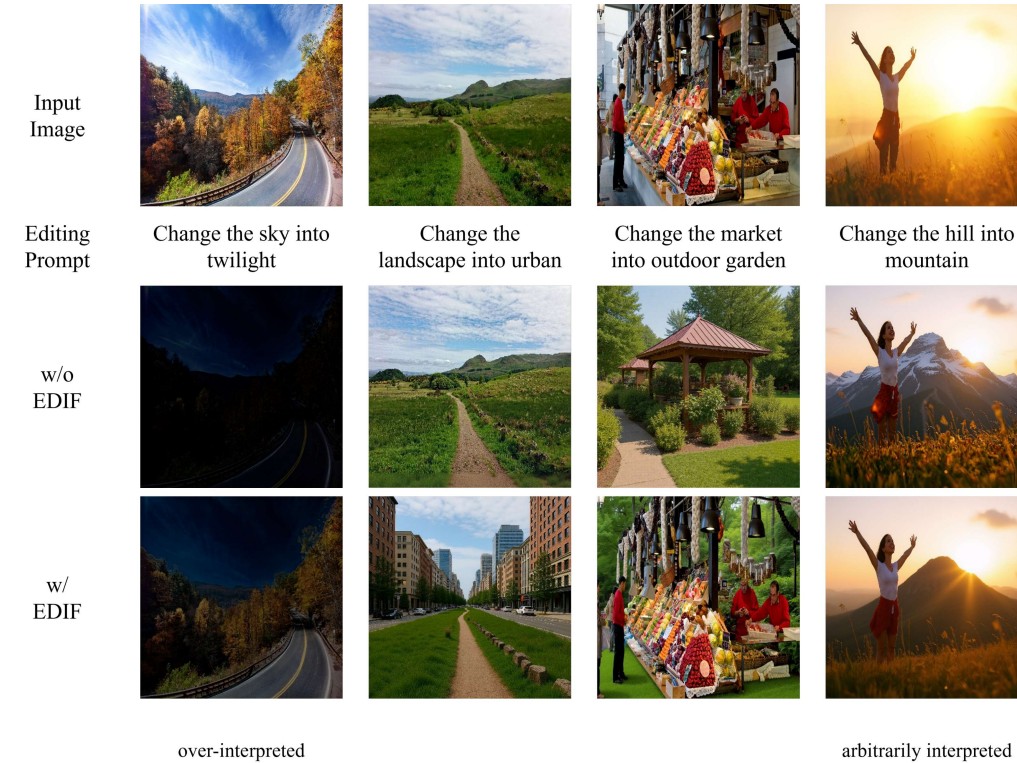

Figure E3: Qualitative results on ImgEdit.

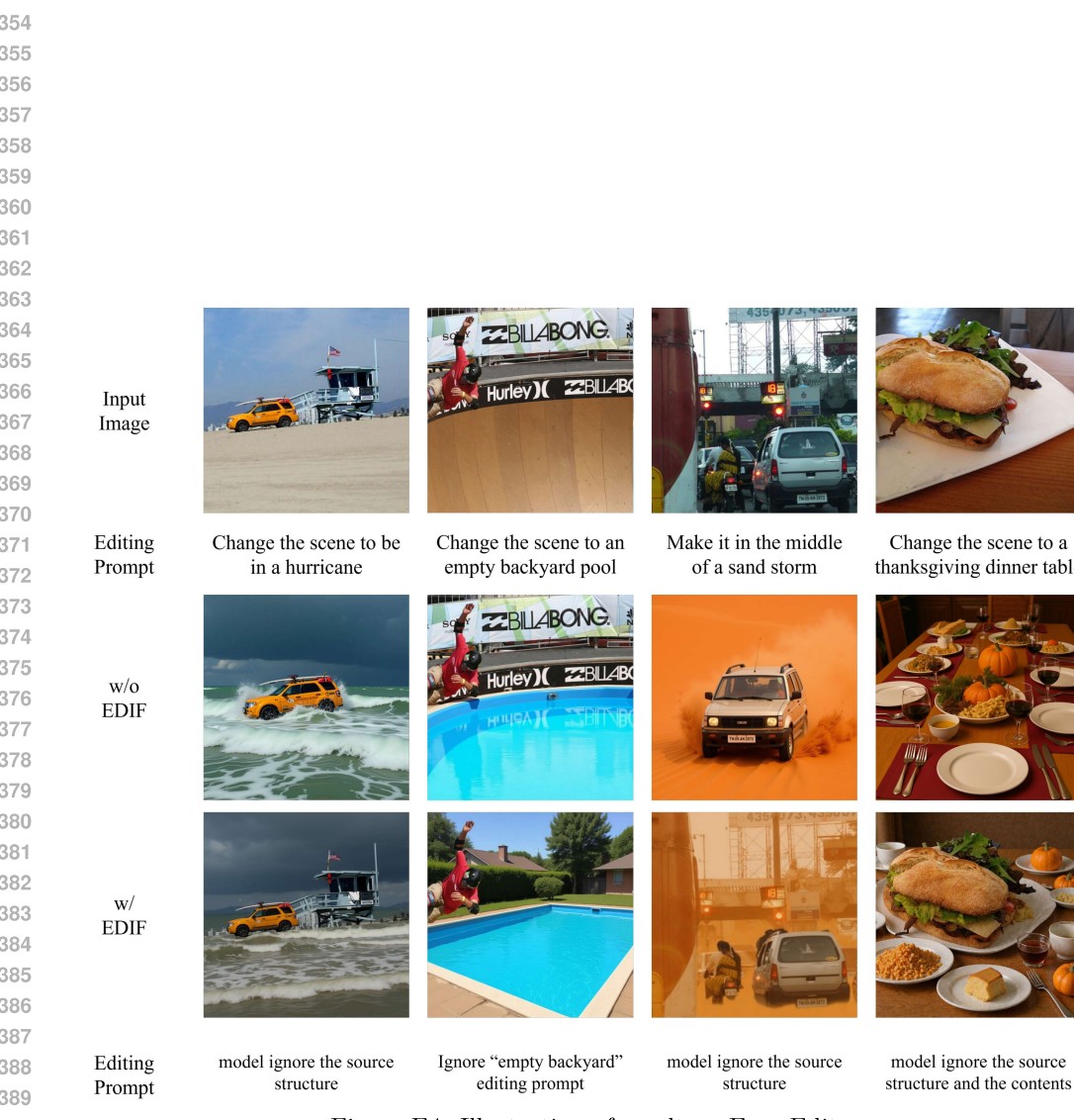

Figure E4: Illustration of result on Emu Edit

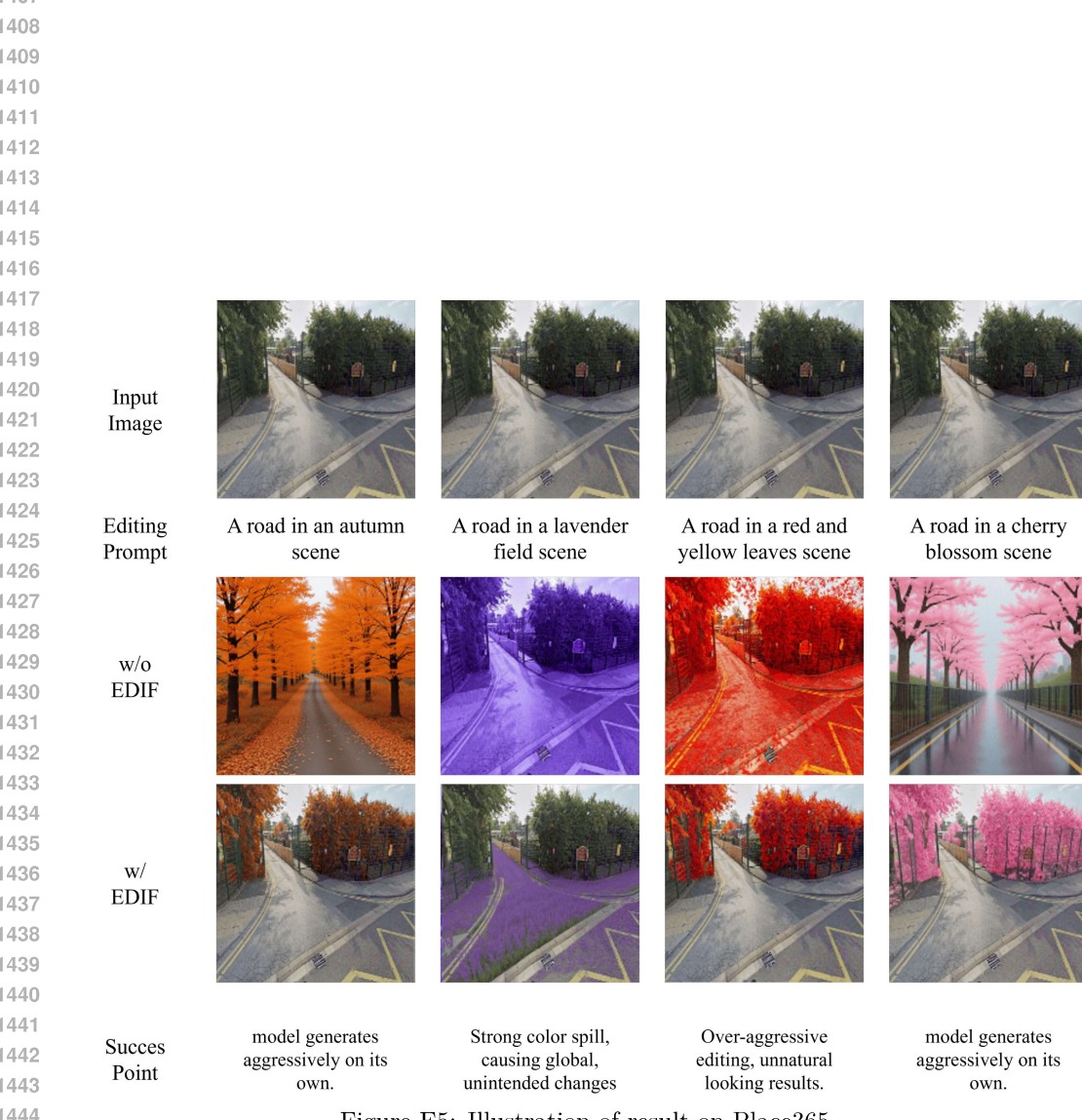

Figure E5: Illustration of result on Place365

|  |  |  |  |  |
|---|---|---|---|---|
| Input Image | | | | |
| Editing Prompt | A café in a living room scene | A café in a lavender field scene | A café in a rustic wooden cabin scene | A café in a cherry blossom scene |
| w/o EDIF | | | | |
| w/ EDIF | | | | |
| Succes Point | In indoor scene editing, the model tends to generate content aggressively, largely ignoring the structure of the original source image. | | | |

Figure E6: Illustration of result on indoor scene.