# OpenReview forum: "EDIF: Editing via Dynamic Interactive Tuning with Feedback"
_ICLR.cc/2026/Conference — Submitted to ICLR 2026_

### Official Review · Reviewer_Pefd · 2025-10-17

**Soundness:** 2
**Presentation:** 1
**Contribution:** 2
**Rating:** 4
**Confidence:** 4

**Summary:**

This paper presents EDIF, a method for scene-centric text-guided image editing that dynamically balances the influence of the source image and the textual editing instruction during diffusion-based generation. Unlike traditional object-centric editing, EDIF targets more complex scene-level edits where global coherence and spatial structure must be preserved. The authors observe that image and text conditions have layer-dependent effects within the diffusion model, and propose to diagnose the editing state via source-image SNR and VLM logits. EDIF constructs a Pareto trade-off curve between fidelity to the original image and adherence to the textual edit, adaptively modulating the denoising process to stay close to this optimal balance.

**Strengths:**

* Introduces an adaptive modulation mechanism using SNR and VLM feedback to control the editing strength during diffusion.

* Provides both empirical and theoretical analysis on layer-wise condition influence and editing diagnostics.

* Experimental results on multiple benchmarks indicate reasonable quantitative and qualitative improvements.

* Conceptually clear in balancing fidelity vs. edit strength via a Pareto trade-off framework.

**Weaknesses:**

* The idea of adaptive conditioning or feedback tuning in diffusion-based editing is conceptually similar to prior guidance or attention-control methods.

* The paper’s focus on “scene-centric editing” is somewhat narrow and not well contextualized with real-world applications.

* Poor presentation quality: Figures are low-resolution, making qualitative evaluation difficult. In addition, formatting inconsistencies and citation errors (e.g., line 53) deviate from the ICLR template, giving the impression of a hastily prepared submission.

* Overall contribution-to-effort ratio is moderate; the framework is incremental rather than groundbreaking.

**Questions:**

* Further clarification on the novelty is appreciated.
* The paper claims that source-image and text conditions have layer-dependent effects. Can the authors provide quantitative or visual evidence (e.g., attention maps, activation statistics) to substantiate this observation?
* The concept of “scene-centric editing” is not clearly defined. What specific criteria distinguish it from conventional object-centric editing, and how does EDIF handle transitions between the two types?
* Since the paper targets practical scene editing, could the authors include comparisons with strong or commercial baselines (e.g., DALLE-3, Firefly, EmuEdit) to demonstrate the competitiveness and real-world applicability of EDIF?

---

> ### Author Response · Authors · 2025-11-21
>
> #### Q1.
> * We clarify that the novelty of EDIF does not lie in introducing a new diffusion architecture, but in uncovering and operationalizing blockwise condition functions that were previously unobserved in DiT-based editing models. Unlike prior guidance or attention-control methods, which apply uniform conditioning across all layers, EDIF demonstrates that image and text conditions exert distinct, layer-dependent influences on structural preservation and editing fidelity. Our key contribution is showing that these functional roles can be systematically profiled and then used as a feedback-controllable mechanism during denoising, enabling stable scene-centric editing without retraining. This capability is not offered by existing editing methods, and to our knowledge, ours is the first work to leverage blockwise condition specialization for Pareto-guided editing. Moreover, although these capabilities exist implicitly in pretrained models, they are often inaccessible in practice because the two condition streams (image and text) frequently misalign, leading to inference failures—even when the base model itself has sufficient representational power. Prior works typically address this mismatch by optimizing the initial noise, or by refining the text embedding, but such approaches operate on only one modality at a time and therefore cannot simultaneously correct both types of misalignment. In contrast, EDIF explicitly diagnoses the evolving editing state and jointly adjusts both image-condition and text-condition pathways, allowing the pretrained model to realize editing behaviors that are otherwise unreachable. This dual-modality correction is the main reason EDIF succeeds in cases where naïve inference or single-path optimization completely fails.
>
> #### Q2.
> * We agree that providing concrete evidence is important. In Appendix C.1, we have already included attention-map analyses that qualitatively illustrate the distinct behaviors of image and text conditions across layers. In the final camera-ready version, we will additionally include the results of our blockwise zeroing experiments, which quantitatively and visually validate the layer-dependent functional roles. These results clearly show that zeroing image-condition or text-condition signals at different blocks produces markedly different editing behaviors, further supporting our claim that the two conditions exert distinct, layer-specific influences.
>
> #### Q3.
> * First, conventional object-centric editing focuses on modifying one or a small number of foreground objects and primarily involves rigid or non-rigid transformations. In this setting, the identity and core structure of the object are preserved while localized modifications—such as changes in color, material, texture, shape, or style—are applied. Most existing text-guided editing methods (e.g., object replacement, object attribute editing, localized attention control) operate under the assumption that the global scene layout remains intact, meaning they are fundamentally designed for object-focused, local editing. Although the term “scene editing” appears in some prior works (e.g., Stable Flow), it is typically used to describe scenarios where a foreground object is present and only the surrounding scene elements are adjusted. In other words, the object remains fixed while the background, lighting, or contextual elements are altered. In contrast, the scene-centric editing we address in this work is a broader and more challenging concept. Scene-centric editing refers to modifying the global semantics of the entire scene—including atmosphere, lighting, background, spatial configuration, environmental attributes, and multi-object relations—regardless of whether a clear foreground object exists. This includes cases where there is no dominant object at all, or where the goal is to transform the meaning of the entire scene rather than adjust a specific object. Because scene-centric editing encompasses a wider and more difficult range of transformations than object-centric editing, existing methods often struggle in this regime. For this reason, we intentionally use the term scene-centric editing to emphasize that our method is designed for global, scene-level transformations rather than localized object-level modifications.
>
> #### Q4.
> * We agree that comparisons with strong or commercial systems such as DALL·E 3, Firefly, and EmuEdit could further illustrate the practical applicability of our method. However, these systems are closed-source, rely on undisclosed inference pipelines, and incorporate prompt filtering and policy interventions, making controlled and reproducible comparisons under identical editing conditions extremely challenging.
> * Since the reviewer has explicitly requested such comparisons, we will provide additional qualitative comparisons with commercial systems to the extent that reproducibility and access allow in the camera-ready version.

---

### Official Review · Reviewer_Tq5x · 2025-10-29

**Soundness:** 4
**Presentation:** 3
**Contribution:** 3
**Rating:** 6
**Confidence:** 4

**Summary:**

This paper proposes a novel text-guided image editing method for scene-centric settings, called Editing via Dynamic Interactive Tuning (EDIF). Unlike existing methods, EDIF constructs a Pareto line between the source image to edited image ratio ($SNR_{src}$) and VLM logits, aiming to strike a balance between preserving the structure of the source image and accurately reflecting the intended edits. The editing process of EDIF is informed by an ablation study that analyzes the block-wise influence of conditions by zeroing out either the image condition or the text condition in specific blocks. This work provides new insights into training-free image editing, and the experiments demonstrate the effectiveness of the proposed method.

**Strengths:**

1. The proposed method is novel and interesting, offering new insights into training-free image editing.

2. The results demonstrate strong performance in scene-centric editing.

**Weaknesses:**

1. Some parts of the presentation are confusing. In line 267, it states, "we first obtain the predicted clean image." Does this imply that EDIF requires one complete editing iteration before adjusting the edits in subsequent iterations? However, other descriptions suggest that EDIF can be completed during denoising iterations. Could the authors clarify this point?

2. The number of adjustments needed during the editing process is not discussed. What is the typical editing time for each image?

3. Some content requires further clarification. The process for constructing an effective Pareto line and determining when the editing is satisfactorily completed is unclear. Additionally, it is not specified which blocks should be adjusted during the editing process.

4. There are typos present, such as a citation error in line 053 and "Pareto frint" in line 249. Some citations also appear incorrect. For example, the citation "Xu et al., 2023a" in line 307 cannot be found in the reference list.

**Questions:**

1. The prompt decomposition transforms free-form prompts into key concept prompts by incorporating keywords such as "add" or "make." However, what about cases where the scene editing only requires the removal of elements?

---

> ### Author Response · Authors · 2025-11-21
>
> #### Q1. Regarding the question on why we state “we first obtain the predicted clean image”
> * Our method performs editing in an iterative manner, and at each step we must evaluate the current editing state to determine how to modulate the image/text conditions. However, directly assessing the current latent is unreliable. As shown in Fig. D1, the latent at intermediate denoising steps is dominated by noise, making the underlying image signal almost invisible. Evaluating such noisy latents would lead to unstable or misleading measurements of both SNR and VLM logits.
> * To address this, we compute the predicted clean image  at each step and use it as the basis for evaluation. The predicted clean image offers a much clearer and semantically meaningful representation of the current state of the denoising trajectory. This allows us to reliably diagnose whether the model is drifting too far from the source structure or under-aligning with the instruction, enabling accurate and stable iterative control. Thus, using the predicted clean image is essential: it ensures that all evaluations are performed on a noise-free approximation of the image, rather than on highly corrupted latents where meaningful comparison is impossible.
>
> #### Q2.
> * We set a maximum of 10 iterative adjustment steps during the editing process. However, in practice the editing loop rarely reaches this upper limit. At each step, we monitor both the SNR and the VLM Logit, and when their changes fall below a small threshold—indicating that the editing state has stabilized—we perform early stopping. As a result, the actual number of adjustment steps per image is typically smaller than 10.
>
> #### Q3.
> * As noted in Appendix C1, we analyzed the effect of zeroing and doubling each block’s image/text conditions. Based on the reviewer’s request, we provide a clearer explanation of how each functional group (G1–G6) was constructed. The final grouping is as follows:
>   * G1 — Lowering image condition improves structural preservation : 5, 6, 33
>   * G5 — Lowering text condition improves structural preservation : 16, 18, 20, 23
>   * G2 — Lowering image condition improves editing fidelity : 5, 6, 16, 17, 18, 19, 20, 24, 43
>   * G6 — Lowering text condition improves editing fidelity : 2, 8, 16, 18, 20, 23, 24, 33, 55
>   * G3 — Increasing image condition improves structural preservation : 0, 29, 31
>   * G4 — Increasing text condition improves editing fidelity : 0, 25, 28, 29, 31, 34
>
> #### Q4.
> * Thank you for the question. The use of verbs such as “add” or “make” in the description of prompt decomposition was intended as an illustrative example rather than a hard restriction. Because our work focuses on scene-centric editing, many of the instructions we consider are phrased as “change the scene into …” or “make it look like …,” which are semantically closer to modification than to pure removal. For this reason, we chose “add/make” as representative verbs in the main text. That said, our decomposition is not limited to additive edits in principle and can be naturally extended to handle removal-type instructions (e.g., “remove cars from the street”), which we will clarify with additional examples in the revised version.

---

> > ### Comment · Reviewer_Tq5x · 2025-11-28
> >
> > 1) Response to Q1 raise a question that how about the performance if other methods edit an image iterativly. Is the comparison to those methods that only edit an image once fair?
> >
> > 2) According to Q1, the proposed method needs to infer the clean image each time. Thus, reporting the average time for each edition is important, which I have asked in Q2.
> >
> > Most of the rebuttal lack clear evidence but say more imformation will be in future version. Therefore, for current version, I will not change my rating.

---

> > > ### Author Response · Authors · 2025-11-28
> > >
> > > #### Q1. Why Iterative Editing Is Essential for DiT-Based Scene Editing
> > > * Our method operates iteratively, as clearly stated in both the paper’s title and the main text. Rather than modifying an image through a single forward pass, our approach progressively adjusts the conditioning based on the current latent state and re-infers the result. This iterative update mechanism is a core component of our method.
> > > * In response to another reviewer’s request, we measured the inference time:
> > >     * Our method (with SNR and VLM Logit computation):~3.78 seconds per denoising step
> > >     * Naive inference (without additional computation):~1.21 seconds
> > > * Thus, our approach introduces approximately 3.12× overhead. When iterative refinement is required, the total runtime naturally increases. However, this iterative process is not merely a speed trade-off—it is essential for enabling scene-level editing in Transformer-based DiT models. Unlike UNet architectures, where the source image can be injected directly into the model via channel-wise concatenation, DiT models rely solely on attention to transmit conditioning. As a result, semantic signals may propagate, but precise spatial alignment is often lost, and naive inference frequently fails to produce meaningful edits. In contrast, our method uses x₀-based feedback at each step to progressively refine the conditioned representation, effectively compensating for the spatial information that DiT architectures inherently struggle to preserve. This allows our approach to succeed in editing scenarios where naive inference consistently fails. Importantly, for simpler editing tasks where strong editing signals naturally emerge in the SNR and VLM Logit estimates, iterative refinement is not required, and a single inference pass is sufficient. While our iterative procedure introduces additional runtime compared to naive inference, it enables editing capabilities that were previously unattainable. We view this as a meaningful and justifiable trade-off.
> > >
> > > #### Q2. x₀-predict
> > > We do not decode the clean image with the VAE at every denoising step. Instead, we simply compute x₀-predict at each step using scheduler. This is explicitly stated in line 267 of the main paper and further clarified in Supplementary Section D.1.

---

### Official Review · Reviewer_WoaN · 2025-10-31

**Soundness:** 2
**Presentation:** 2
**Contribution:** 2
**Rating:** 0
**Confidence:** 5

**Summary:**

This paper does not follow ICLR's official template. I suggest a desk rejection.

**Strengths:**

/

**Weaknesses:**

/

**Questions:**

/

---

> ### Author Response · Authors · 2025-11-21
>
> * Thank you for your comment regarding the formatting. We have carefully reviewed the manuscript, and we did not find any components that deviate from the official ICLR LaTeX template. The structure, style files, and formatting rules appear to be fully aligned with the required template. Nevertheless, to avoid any potential misunderstanding, we will thoroughly re-verify the formatting once more and ensure that the camera-ready version strictly adheres to all official guidelines. We kindly ask that the paper be evaluated based on its technical contributions, as the current version follows the ICLR template to the best of our knowledge.

---

### Official Review · Reviewer_5cbW · 2025-11-06

**Soundness:** 2
**Presentation:** 2
**Contribution:** 2
**Rating:** 4
**Confidence:** 5

**Summary:**

This paper focused on scene-centric editing setting, and proposed Editing via Dynamic Interactive Tuning (EDIT) to achieve adaptive trade-off between source-image structure and instruction fidelity. This paper pointed out the block-wise variation inside the diffusion models, i.e., both the image condition and text condition functions independently. This paper used the signal-to-noise ratio and VLM logits to diagnose the editing state, and then using them to adaptively modulate the source-image and editing-text condition to achieve balanced editing results.

**Strengths:**

1. This paper focused on the scene-centric editing, which is more challenge compared with the object-centric editing in existing works and is an important research direction.
2. This work proposes a reasonable method to achieve dual optimization of source preservation and prompt fidelity

**Weaknesses:**

1. The writing and organization of this paper are inadequate and require substantial revision. There’re many typos such as the “?” citation in line 53.
2. The method is plug-and-play, but the fact that it was only tested on Kontext weakens the generalizability of the study. Can this method be used on other image editing base models besides Kontext?
3. The quantitative experimental results of this work did not show significant improvement.
4. User studies need to include more users and explain the specific rating criteria.

**Questions:**

See weakness

---

> ### Author Response · Authors · 2025-11-21
>
> #### Q1.
> * Thank you for pointing out the incorrect citation. I have corrected it.
>
> #### Q2. Application on other method?
> * Our method requires the ability to independently control image and text conditions; thus, it cannot be directly applied to U-Net–based editors where image conditioning is injected through channel-wise concatenation and cannot be separated from the latent. To test generalizability beyond Kontext, we also conducted zeroing analysis on Qwen-Image Editing, another DiT-based model. Unlike Kontext, Qwen did not show clear blockwise functional patterns, likely because it mixes source-image description and editing instruction within the text branch, breaking the independence between image and text conditions. Even so, zeroing experiments on Qwen revealed distinguishable block groups—some affecting source preservation and others affecting editing fidelity. Based on these observations, we applied EDIF to Qwen by clustering blocks with similar behaviors rather than using the six groups (G1–G6) defined for Kontext. We identified each group’s role by observing SNR and VLM-logit changes after initial adjustments, and then performed dynamic condition control accordingly. The experimental results were highly encouraging. Below, we provide results obtained on the HQ dataset.
> | Method          | Sampling         | CLIP Img | CLIP out | L2 |
> |-----------------|------------------|----------|----------|-----------|
> | Qwen-ImageEdit  | Fixed Condition  | 0.913    | 0.28     | 0.040  |
> | Qwen-ImageEdit  | Edif             | 0.931    | 0.293    | 0.030  |
> * To further quantify the degree of source-image preservation, we additionally report the L2 score (following the evaluation protocol of “InitNO: Boosting Text-to-Image Diffusion Models via Initial Noise Optimization”). Although Qwen-ImageEdit did not exhibit as clear functional blockwise patterns as Kontext, our results show that adjusting the conditioning still leads to improved performance. As summarized in the table, applying EDIF consistently enhances both editing fidelity and source-preservation metrics compared to the fixed-condition baseline.
>
> #### Q3. The quantitative experimental results of this work did not show significant improvement.
> * The primary goal of this work is not to maximize a single metric, but to provide interpretability and controllable editing dynamics that prior methods lack. Although average numerical gains may appear small, EDIF succeeds on many cases that were previously “uneditable,” demonstrating meaningful practical improvement. Unlike existing scene-centric approaches that require costly retraining, EDIF is a training-free, inference-only framework that resolves failure modes unreachable by fine-tuning. Prior training-free techniques such as noise or condition optimization work well for T2I, but are insufficient for TIE, which must satisfy both source preservation and instruction fidelity simultaneously. EDIF overcomes this by controlling image/text conditions inside the pretrained model, diagnosing the editing state using $\mathrm{SNR}_{\text{src}}$ and VLM logits, and adjusting conditions based on our blockwise analysis. This feedback-driven control enables a balanced and reliable editing trajectory without additional training.
>
> #### Q4. User study
> * Thank you for raising the concern regarding the scale of the user study and the clarity of the evaluation criteria. We address your comments as follows. First, our user study was not intended as a broad preference survey, but rather as a focused expert evaluation on challenging scene-centric editing cases. We intentionally curated difficult and ambiguous samples from three datasets—ImgEdit-Bench, Emu Edit Bench, and Places365—excluding trivial cases where all models perform similarly. As a result, each trial required participants to carefully judge fine-grained differences in structure preservation and instruction fidelity.
> * Second, we recruited 30 participants, consisting primarily of AI researchers, designers, and practitioners with substantial experience in image generation and editing. Unlike general crowd-sourced evaluations, our study was conducted by users who can critically and reliably assess structural consistency, prompt alignment, and visual realism. This provides a much more rigorous and meaningful evaluation than a large but unfiltered user pool.
> * Finally, we agree with the reviewer’s suggestion and will clarify these criteria more explicitly in the final version. We are also open to extending the user study and providing supplemental analyses if needed.

---

### Official Review · Reviewer_W2As · 2025-11-06

**Soundness:** 2
**Presentation:** 2
**Contribution:** 2
**Rating:** 4
**Confidence:** 5

**Summary:**

This paper proposes EDIF to address the trade-off between source-image structure preservation and instruction fidelity in scene-centric text-guided image editing. The author shows that the source image condition and the embedding act with layer-dependent directions. Therefore, this paper uses source SNR and VLM logits to diagnose the editing state. Extensive experiments on three benchmarks demonstrate the effectiveness of EDIF.

**Strengths:**

1. This work is clearly expressed and easy to understand.
2. This work provides extensive experimental results and comprehensive comparisons with multiple baselines.

**Weaknesses:**

1. The increased time consumption of EDIF compared to baseline was not mentioned.
2. The function differences between different blocks are largely based on empirical observations, which lack theoretical support. Can these function differences be extended to other image editing methods?

**Questions:**

N/A

---

> ### Author Response · Authors · 2025-11-21
> **Runtime Analysis and Application on other domain**
>
> #### Q1. increased time consumption
> * In our method, each diffusion denoising timestep requires approximately 3.78 seconds when computing both the SNR and the VLM Logit at inference time. In contrast, a naïve inference without SNR and VLM Logit computations takes 1.21 seconds. Thus, our approach introduces roughly (3.78 / 1.21 ≈ 3.12×) runtime overhead per denoising step. If the editing process requires additional iterative refinement, the total runtime may increase further. However, omitting our method and relying solely on naïve inference often makes meaningful editing impossible. Across numerous cases especially on real images like the EmuEdit dataset, we observed that seed control (the initial noise control) or timestep adjustment, or manipulation of the text-condition guidance scale never produce successful edits.
> * One might argue for retraining the entire model as an alternative. Yet this solution is rarely practical: retraining must be repeated whenever the data distribution, editing style, or application domain changes, and preparing a new training set—including curation, annotation, filtering, and quality assurance—requires substantial time and resources. Moreover, large-scale diffusion model training itself is computationally expensive and operationally burdensome. Even worse, a newly trained model is not guaranteed to generalize; it can easily fail again when exposed to a different dataset or a slightly shifted real-world distribution, forcing yet another cycle of retraining.
> * Therefore, although our method introduces some additional inference-time cost, it enables edits that are fundamentally unattainable with naïve inference. For challenging real-image editing, this trade-off is both justified and highly valuable. In short, our approach offers a practical and effective solution where standard inference completely fails, making the added runtime a meaningful and worthwhile investment.
>
> #### Q2. theoretical analysis of blockwise function or application to other domain
> * Block-level functions may not directly transfer to other domains in a plug-and-play manner, but analyzing block-specific characteristics and exploiting layerwise patterns are well-established research directions across many areas of deep learning. Prior studies consistently report that different Transformer blocks play distinct functional roles. For example, pruning and distillation methods often remove blocks with lower functional contribution, and StableFlow identifies block-dependent functional patterns that can be selectively controlled attention. From this perspective, our blockwise analysis is aligned with—and contributes to—the broader line of work that leverages layerwise specialization to achieve more controllable and efficient model behavior.

---

### Meta-Review · Area_Chair_vZGz · 2025-12-29

**Summary:**

The paper initially received three negative and one positive ratings. The concerns are mostly about 1) computation and fairness of the iterative editing scheme, 2) insight and choice of adjusting some DiT blocks, 3) generalization to other editing methods, 4) more comparisons and performance gains against existing methods.

**Reviewer Concerns:**

The authors have provided responses in the rebuttal to answer initial concerns from the reviewers. Most responses are explanatory ones, except for one additional experiment to use the Qwen-Image Editing model with the proposed method. The AC took a close look at the paper, reviews, and the rebuttal. After the rebuttal, the AC finds that most questions (as summarized in the above "Summary" section) are not well addressed. Especially, even though there is one more result using Qwen-Image Editing, it shows that the proposed method is not directly applicable to make the performance better, which also reflects reviewers' opinion (W2As and Tq5x) about the empirical choice of adjusting DiT blocks. Moreover, considering the problem scope, performance gain, and its additional computational requirement, the AC agrees with the reviewers' overall feedback and hence recommends the rejection rating.

**Reviewer Scores:**

Reviewer Tq5x mentioned to retain the original rating, while the other three reviewers did not fully participate the discussion.

---

### Decision · Program_Chairs · 2026-01-26

Reject